# A Secure and Lightweight Group Mobility Authentication Scheme for 6LoWPAN Networks

**DOI:** 10.3390/s25051458

**Published:** 2025-02-27

**Authors:** Fatma Foad Ashrif, Elankovan A. Sundararajan, Mohammad Kamrul Hasan, Rami Ahmad

**Affiliations:** 1Research Centre for Software Technology and Management, Faculty of Information Science and Technology, Universiti Kebangsaan Malaysia, Bangi 43600, Selangor, Malaysia; elan@ukm.edu.my; 2Computer Science Department, Sebha University, 5 October Street, Sebha 18758, Libya; 3Research Center for Cyber Security, Faculty of Information Science and Technology, Universiti Kebangsaan Malaysia, Bangi 43600, Selangor, Malaysia; mkhasan@ukm.edu.my; 4College of Computer Information Technology, American University in the Emirates, Dubai 503000, United Arab Emirates; rami.alshwaiyat@aue.ae

**Keywords:** PMIPV6, group handover, secure authentication, 6LoWPAN

## Abstract

The integration of Internet Protocol version 6 over Low-Power Wireless Personal Area Networks (6LoWPANs) provided IP technologies within wireless sensor networks that dramatically increased the Internet of Things (IoT). Therefore, to facilitate efficient mobility management for resource-constrained IP-based sensor nodes, the Proxy Mobile IPv6 (PMIPv6) standard has been introduced to reduce communication overhead. However, the standard has addressed security and mobility authentication challenges in 6LoWPANs, although recent solutions have yet to focus much on facilitating secure group handovers. Considering these issues, a Secure and Lightweight Group Mobility Authentication Scheme (SL_GAS) is proposed for 6LoWPAN’s highly constrained sensor nodes. SL_GAS innovatively utilizes one-time alias identities, temporary IDs, tickets, and an aggregated MAC with tags to ensure mutual authentication while maintaining sensor anonymity, providing a balanced security and privacy approach. SL_GAS’s robustness against a variety of security threats is validated through formal automated verification using the Scyther tool alongside SVO logic, while an informal analysis demonstrates its resilience to known attacks. Comparative analysis with existing schemes highlights SL_GAS’s advantages in reducing signal cost, transmission delay, communication, and computation overhead. SL_GAS stands out for its combination of security, privacy, and efficiency, making it a promising approach for enhancing IoT connectivity in resource-constrained settings.

## 1. Introduction

Introducing the Low-Power Wireless Personal Area Network (6LoWPAN) has significantly increased the scalability of the Internet of Things (IoT) infrastructure. As a result of its ability to effectively integrate IPv6 over low-power wireless networks, 6LoWPAN has enabled widespread connectivity and enhanced communication among a variety of IoT devices [1]. With this advancement, sensor networks will be deployed more extensively and diversely, facilitating more incredible innovation and expansion in the IoT sector. It is estimated that 29 billion IoT devices will be deployed by 2030 due to the IoT communication used in various industries, including healthcare, agriculture, manufacturing, and transportation technologies [2]. More than 80 companies are standardizing a protocol on a 6LoWPAN layer called “Thread”, which allows smart home devices to be connected and controlled optimally [3]. 6LoWPAN sensor nodes are typically resource-constrained devices with limited processing power, memory, and energy capacity. While 6LoWPAN nodes can feasibly perform lightweight symmetric cryptographic operations such as Advanced Encryption Standard (AES) and Hash-Based Message Authentication Codes (HMACs), more computationally intensive asymmetric cryptographic operations, like Elliptic Curve Cryptography (ECC), are challenging to implement directly on these devices [4]. To address these limitations, cryptographic tasks can be offloaded to intermediary entities, such as access gateways or authentication servers, ensuring secure communication while preserving the nodes’ energy and computational resources. This heterogeneity in resource capabilities necessitates lightweight and efficient security solutions designed specifically for the 6LoWPAN environment [1]. The 6LoWPAN-based wireless sensor network (WSN) has dramatically expanded the application space of WSNs, and an increased number of mobile devices are connected to the Internet, so seamless Internet connectivity is increasingly important [5]. IoT ecosystems are rapidly adopting mobile devices, which increases the demand for secure, lightweight, and scalable group authentication protocols [6]. In 6LoWPAN networks, secure communications for mobile devices are further complicated by frequent handovers, resource constraints, and the need for mobility-specific protection. Mobile IPv6 (MIPv6) was developed to ensure seamless communication for mobile devices by maintaining uninterrupted network connectivity as devices move across domains [7]. Building upon this foundation, the Internet Engineering Task Force (IETF) introduced Proxy PMIPv6, a network-based localized mobility management protocol designed to reduce signaling overhead, particularly for resource-constrained 6LoWPAN nodes. Despite these challenges, PMIPv6 enables resource-constrained devices to function efficiently by delegating cryptographic tasks to access gateways and reducing mobility-related signaling overhead. PMIPv6 reduces signaling overhead by delegating cryptographic and mobility-related tasks to access gateways, enabling efficient operation of low-power nodes [8]. However, despite its benefits, PMIPv6 still faces challenges in ensuring secure and scalable group authentication, particularly during group handovers, where multiple devices transition across domains simultaneously. Existing solutions often fail to provide robust protection against key leakage, ensure Perfect Forward Secrecy (PFS), or resist various advanced threats. These challenges highlight the need for an advanced, secure, and lightweight authentication scheme that can address the unique requirements of 6LoWPAN networks while ensuring efficient and secure group mobility.

### 1.1. Motivation

PMIPv6 faces critical challenges in securing group mobility. Frequent handovers of mobile nodes in 6LoWPAN networks introduce vulnerabilities such as replay attacks, impersonation, and denial-of-service (DoS) attacks, which can exploit insecure handovers. Additionally, traditional authentication mechanisms such as ECC are too computationally expensive for 6LoWPAN nodes, and existing solutions often prioritize individual handovers, failing to address group mobility scenarios where multiple nodes transition adequately simultaneously [6,9]. Furthermore, many current protocols neglect essential cryptographic guarantees, including Perfect Forward Secrecy (PFS), resistance to key leakage, and protection against tracking. These gaps underscore the urgent need for an advanced solution that addresses these challenges while remaining compatible with the strict resource constraints of 6LoWPAN devices. As IoT ecosystems expand, the demand for scalable and secure group authentication protocols has grown significantly. Current proposals for group mobility authentication schemes provide partial security for PMIPv6 networks but often lack critical features [10,11,12,13]. They fail to ensure Perfect Forward Secrecy (PFS), leaving past communications vulnerable if session keys are compromised. Additionally, most existing protocols do not adequately safeguard against adversaries tracking the movement of devices across domains or protect against key leakage and DoS attacks. These limitations highlight the necessity of a secure and lightweight group authentication scheme that can cater to the needs of 6LoWPAN networks while effectively addressing advanced security requirements.

### 1.2. Contribution

The following summarizes our contribution:Secure and Lightweight Group Mobility Authentication Scheme (SL_GAS) proposed using secret parameters in the registration phase to ensure that the session keys are confidential and robust, providing a robust foundation for secure communication against attacks such as replay attacks and ensuring that each session’s parameters are unique.SL_GAS proposed a ticket during the initial group authentication phase for all mobile nodes and the group leader, enhancing the group authentication process’s security and enforcing access control in dynamic and resource-constrained environments. It provides a secure and efficient way to manage authenticated sessions, prevent attacks, such as DoS and key leakage, and provide PFS and untraceability. Additionally, it reduces the overhead associated with authentication, provides continuity and resumption of sessions, and optimizes resource utilization that does not require key redistribution.SL_GAS used aggregated MAC with a tag approach that employs an improved authentication efficiency and integrity verification while minimizing computational overhead, unlike existing schemes that rely on plaintext MAC transmission.We formally analyze SL_GAS, considering Dolev–Yao (DY) [14] and Canetti–Krawczyk (CK) and extended CK [15] intruder adversary models, and informally analyze utilizing SVO logic and the Scyther tool (v1.1.3, Cas Cremers, Saarbrücken, Germany, and Oxford, UK). SL_GAS can provide additional security properties and the ability to withstand various protocol attacks.Performance evaluations have demonstrated that SL_GAS handover authentication has an advantage concerning signal cost, transmission delay, and computation overhead.

### 1.3. Paper Outline

The following section discusses the structure of the remainder of the paper: Section 2 discusses related research. Section 3 outlines our system model, requirements, threat model, and design objectives. Section 4 details our proposed protocol. Section 5 provides the formal and informal security analysis of SL_GAS. Section 6 discusses the performance evaluation and provides a discussion. The conclusion is presented in Section 7.

## 2. Related Works

A brief overview of the literature on group handover authentication schemes for 6LoWPAN networks is provided in this section. In addition, summarize and compare SL_GAS with the most relevant schemes in Table 1. Several solutions have been proposed for reducing handover delays and packet losses during single-node handovers [11,12,16]. However, little focus is given to security issues that could cause severe problems for resource-constrained IP-based mobile nodes. Chen et al. [17] provided a group mobility protocol that was designed for wireless body area network (WBAN)-based 6LoWPAN to support many devices, reduce signaling messages, and reduce the execution time to switch devices. The performance of handovers in 6LoWPAN networks has improved in regard to various aspects, but security issues have not been addressed much. Wang and Mu [18] presented a secure mobility protocol to protect the mobility of devices in the 6LoWPAN network. However, the paper does not elaborate on the procedure for pre-distributing the pairwise key within the nodes. A secure password authentication mechanism (SPAM) [19] is proposed for fast and secure handovers in PMIPv6. However, whenever a mobile node tries to connect to another gateway, the SPAM requires full authentication, resulting in a long delay during handover. The security of the group handover for 6LoWPAN is still insufficient, with current solutions only supporting secure handover for individual nodes.

Therefore, the SGMS (Secure Group Mobility Scheme) was proposed in [11]. A cryptographic algorithm is used in the SGMS to ensure simultaneous handovers of multiple nodes. Despite its efficiency, this scheme still has security issues, such as key leakage attacks and PF/BS. A secure and seamless IP (SEIP) protocol for group lightweight authentication was proposed in [16]. Symmetric cryptosystems are employed to minimize computational overhead. SEIP resists redirection and DoS attacks [16]. However, it does not resist replay, MITM attacks, and PFS/BS, making it unsuitable for highly dynamic mobile environments. Imran et al. [20] proposed a scheme named “CBAS”. Its primary goal is facilitating efficient and secure group authentication in 6LoWPAN networks. The performance evaluation of CBAS demonstrates that it has effectively decreased the registration time, signaling cost, and handoff authentication delay compared to the SGMS. However, it does not provide PFS, whereas key management weaknesses may be compromised during the handover process to compromise the integrity of session keys, thereby challenging the PFS. In addition, due to authentication overhead, devices may not be able to successfully transition to a new network, effectively causing DoS attacks. Our analysis shows that there are a few solutions to target group security in PMIPv6-based 6LoWPANs. Despite significant interest in the 3rd Generation Partnership Project (3GPP), there has been a lack of focus on security issues [3]. Moreover, in the schemes that use the aggregated MAC technique, the identities of each mobile node are transmitted in plain text over an insecure channel and, therefore, cannot protect the privacy of the node. Additionally, these schemes are vulnerable to DoS attacks since all members must be legal for the MAC to be successfully verified. The adversary can deliberately provide fake MAC addresses to the group leader, resulting in the intentional failure of the entire group verification process. It is important to balance security, privacy, and efficiency in IoT and mobile networks. Authentication schemes that rely on static identities have historically been vulnerable to tracking attacks, replay attacks, and identity theft [1]. Several studies [4,5] have addressed this issue by utilizing one-time user identities, ensuring that each session uses a fresh, unlinkable identity each time. In resource-constrained environments such as 6LoWPAN-based IoT networks, alias-based authentication enhances privacy while maintaining strong security guarantees. Similar to conventional authentication schemes that require individual message authentication, aggregated MAC-based authentication has been proposed as an efficient alternative. The use of aggregated MACs allows multiple authentication messages to be verified simultaneously, which reduces processing overhead and increases scalability [6]. Consequently, a secure and lightweight group authentication scheme (SL_GAS) is proposed for highly constrained sensor nodes in PMIPv6-based 6LoWPAN. By contrast with existing schemes, SL_GAS uses both formal and informal security analysis techniques, such as Scyther, SVO logic, and adversary models (DY, CK, and eCK), to validate the system’s security properties. Through these evaluations, we ensure that our systems are resistant to impersonation, replay, MITM, and privileged insider attacks. We also guarantee that our systems are untraceable and secure. Using ticket-based authentication, as well as secret parameters, one-time alias identities, temporary IDs, and an aggregated MAC with tags, providing superior security and efficiency for highly constrained PMIPv6-based 6LoWPAN networks.

## 3. System Background

A description of the network model and threat model, as well as the design goals that are considered, is presented in this section.

### 3.1. Network Model

As shown in Figure 1, the components of the network model are explained as follows:The 6LoWPAN mobile nodes (6L_MNi_): There are two categories of 6L_MNi_, which are portable devices utilized for gathering sensory data. Full Function Devices (FFDs) are devices with full functions, while Reduced Functions Devices (RFDs) have limited functions. There is sufficient power and memory in FFDs to enable them to be used as routers, while RFDs cannot serve as routers. Whenever a cluster of 6L_MNi_ is in operation, the supervisory node or group leader (6L_MNGL_) must be an FFD responsible for managing. Mobility-related messages are sent between the mobile access gateways and the supervisory node. All 6L_MNi_ nodes in a 6LoWPAN network can move as a group across multiple domains.

**Figure 1 sensors-25-01458-f001:**
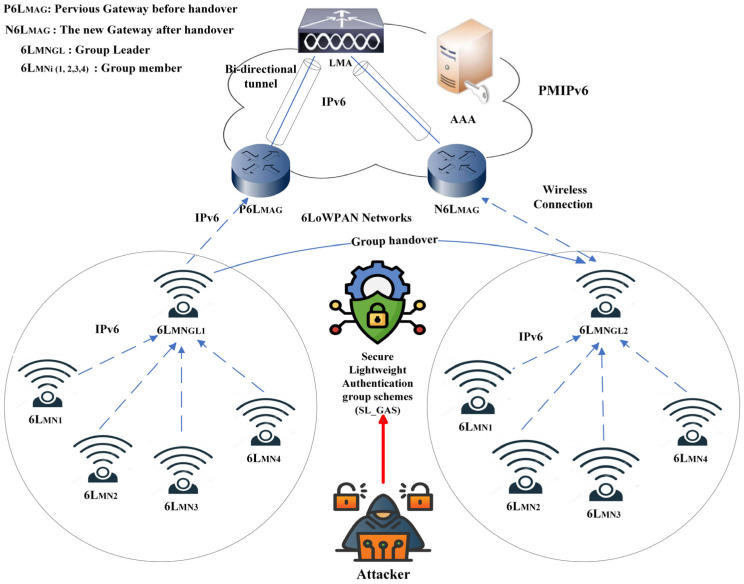
A group handover transitioning from one gateway to another in a 6LoWPAN-based PMIPv6.

The 6LoWPAN mobile access gateways (6L_MAGs_): It is 6L_MAGs_’ responsibility to track the movement of the 6L_MNi_ to transmit mobility-related signals to the local mobility anchor (LMA). Data packets are transmitted between 6L_MAGs_ and an LMA using a bidirectional tunnel. The hierarchical architecture of 6L_MAGs_ ensures that a compromised 6L_MAGs_ affects only its connected nodes. The integrity of the data is protected by integrity checks and mutual authentication.LMA: Among its responsibilities is maintaining the binding states of all 6L_MNi_ currently attached to the router; in addition, every registered node is assigned a unique network prefix. This enables a 6L_MNi_ to configure its IP address by its needs. LMA is used to route messages intended for these addresses. Deploying redundant LMA instances with secure failover mechanisms is essential to ensure network resilience. This strategy allows backup LMAs to maintain network operations seamlessly in the event of a compromise. Regularly updating cryptographic keys and employing secure storage techniques minimize potential attacks’ impact by limiting the window of opportunity for exploitation. Additionally, protecting communication channels between the LMA and other network entities with secure protocols, such as IPsec or TLS, ensures transmitted messages’ confidentiality, integrity, and authenticity.The accounting, authorization, and authentication server (AAA): AAA’s important function is to identify and authenticate 6LMNi users and authorize their access to the network. Policy files of the 6L_MNi_ are stored in AAA as configuration information. A wired link is established between the 6L_MAGs_, LMA, and AAA using high-speed copper cables with a low bit error rate. Signaling messages are protected by Internet Protocol Security (IPsec), which offers authentication, confidentiality, and integrity of data, as well as other network security services [21]. Establishing secure associations using the Internet Key Exchange Protocol version 2 (IKEv2) to pre-share the symmetric keys between different parties is also possible. Despite IKEv2 and IPsec being mandatory with IPv6, they may not be appropriate for 6L_MNi_ due to their insufficient capabilities.

### 3.2. Threat Model

SL_GAS was designed based on the DY, CK, and eCK adversary models, which are summarized in Figure 2 based on the key security attributes considered. Despite insecure communication channels, all models differ in terms of adversary query capability.

The DY threat model considers communication parties to be honest and allows them to operate multiple sessions simultaneously [4]. There is a high level of security throughout the communication channel, and an adversary can capture, redirect, delete, rearrange, replay, and program the communication schedules. The adversary can act as the attack director for MITM attacks, resulting in various MITM techniques. The security attributes are PFS, key confirmation, key independence, and session state compromise. The CK and eCK models are widely accepted for AKE schemes. This adversary model allows attackers to access a session’s secret randomness by compromising a pseudo-random number generator. The CK model considers PFS, key confirmation, key independence, and compromise of the session state. The eCK model considers ESL, key compromise impersonation, and session state compromise. An adversary may also be able to compromise the session and access it. An attacker can access long-term keys. The eCK model is inherently more vulnerable to ESL attacks than the CK model due to its major disadvantage: the adversary can access ephemeral secrets [22]. As a result of the leak of ephemeral keys, it appears that the attacker is capable of determining the randomness generated by the generator with high probability. Specifically, SL_GAS is designed to secure the 6LoWPAN against the following attacks:

Replay attack: A malicious actor intercepts legitimate packets and retransmits them at a new time or place, typically intending to cause resource depletion. A replay protection mechanism between LMA, 6L_MAG_, and 6L_MNi_ (e.g., timestamps and nonces) detects and blocks malicious behavior on the channel when integrity checks (e.g., HMAC) are violated or compromised.Impersonation attacks involve adversaries attempting to assume the identities of access and mobility functions, 6L_MNi_, 6L_MAG_, and AAA, to deliver packets using their credentials.Unauthorized access refers to the attempt made by external attackers, who are not authorized node 6L_MNi_, to gain access to the network. Intra-group messages are encrypted at the group level to prevent unauthorized access.Session key leak and PFS: Ephemeral keys ensure each session’s security and independence. However, if these keys are leaked, an attacker can decrypt communications, impersonate nodes within the session, and compromise forward secrecy. Ephemeral keys provide forward secrecy as one of their primary purposes, meaning that capturing these keys allows attackers to decrypt past communications, thus undermining both session-specific security and the overall integrity of the communication system. AAA and destination 6L_MNi_ establish a shared key during handover protocols. With this key, a malicious 6L_MAG_ entity may attempt to retrieve the past keys employed by the AAA and those utilized in forthcoming communications.MITM attack: A malicious node attempts to create a shared key with the AAA, LAM, 6L_MAG_, and 6L_MNi_. This allows the attacker to manipulate and decrypt the messages being sent.DoS attack involves an assailant engaging in malicious actions to obstruct a legitimate user’s access to network resources.A privacy-preserving authentication system safeguards the confidentiality of all device information during the authentication process by employing privacy protection measures.

### 3.3. Performance Goals

To achieve efficient and secure group handover authentication for PMIPv6, the design of security goals for SL_GAS should include the following:Traceability and anonymity: 6L_MN_ should update its anonymous identification during inter-domain handover, which can only be known by AAA.Perfect Forward Secrecy (PFS): the adversary must be unable to extract any valid information from the previous ciphertexts when the current key is compromised.Mutual authentication and key agreement (AKE): AAA verifies the group’s identity during the initial authentication phase and informs LMA and 6L_MAGs_ of the results. The group must simultaneously authenticate the identities of 6L_MAGs_, LMA, and AAA to increase security. It is essential to confirm the legitimacy of both the group and the target 6L_MAGs_ before the group proceeds with its operations. Establishing a secure session key between the group leader, 6L_MNGL_, and the 6L_MAGS_ is also necessary to ensure the confidentiality of the ensuing data transmission during the initial authentication and handover authentication.Protocol attack resistance: For the designed scheme to be secure and effective, it must be resilient to attacks, as explained in the treat model section, including replay, impersonation, MITM, DoS, Sybil attacks, etc.Performance optimization: SL_GAS must consider computation overhead, signal cost, and transmission overheads to reduce the authentication delay. Thus, the performance of SL_GAS for initial authentication and handover authentication must be superior to that of existing schemes.

## 4. System Models

Models and preliminary results used in SL_GAS are provided in this section.

### 4.1. Aggregated Message Authentication Codes (Agg-MAC)

**Definition** **1.**
*MAC aggregates [23] are a set of probabilistic polynomial times (PPTs) and are explained as follows.*

*MAC: Let κ ∈ {0,1}λ be a key with a length equal to the security parameter, λ, and let msg ∈ {0,1}∗ be any arbitrary length message; MAC is constructed from the standard keyed pseudo-random function, tag ← MACκ(msg) = Fκ(msg), where F is a pseudo-random function.*

*Agg-MAC: Let (msgi, di) and tagi be a message/identifier pair and its corresponding tag for node i, respectively. The new message (M, tag) is the aggregation of all messages, i.e., M = {(msg1, id1), (msg2, id2), (msg3, id3), …, (msgl, idl)}, and the corresponding tag is constructed by simply XOR-ing all the messages: tags, tag = tag1 ⊕ tag2 ⊕ tag3 ⊕ ⋯ ⊕ tagl.*

*Verify: Given the set of all identifier keys {K1, K2, K3, …, Kl}, and the message tag pair, (M, tag), the verify algorithm, verify (K1, ID1), (K2, ID2), …, (Kl, IDl), (M, tag), outputs upon successful authentication; and it is zero otherwise.*



### 4.2. LiCi Block Cipher

LiCi block cipher: LiCi [24] uses a 64-bit plaintext encryption key and 128-bit plaintext encryption. In comparison with other ciphers, LiCi consumes only 1051 gate equivalents. A Virtex 6 FPGA makes ciphers that consume 25 mW of power. LiCi is an excellent option for applications requiring a low footprint and low power among the many ciphers available. The cipher has a key-scheduling feature. LiCi utilizes the PRESENT key schedule as an AES algorithm. A LiCi cipher goes through 31 iterations or rounds until the optimum security level is reached. LiCi has developed a lightweight S-box with a shift mechanism [25].

### 4.3. The Proposed Scheme (SL_GAS)

6LoWPAN devices generate IPv6 interface identifiers (IIDs) instead of IEEE media access control addresses using the method described in [26]. This IPv6 stateless address auto-configuration technique makes it possible to reduce the risks associated with IPv6 address tracking and address scanning, leading to a more private and secure way of generating IIDs. The 6L_MAGs_ are connected to the AAA via wireless communication via the LMA. A group of 6LMNs in the network must register with the AAA via a secure channel to establish a trust connection with the LMA. The use of tickets provides dynamic, time-limited, and cryptographical protection, which is suitable for scenarios requiring a high level of security. By including a Group ID (GID), a ticket can combine the advantages of both approaches. The simplicity of GIDs can be balanced with the cryptographic protection of tickets and their dynamic properties. Using secret parameters in the registration phase that are only known by AAA increases security because they can act as a unique identifier. As long as the secret parameter is changed in each session, it would enhance security by reducing the risks associated with long-term exposure. The aggregation MAC with tag ensures the mutual authenticity and integrity of the 6L_MN_. SL_GAS is designed in three phases: registration, initial group authentication, and group handover authentication. The notations and descriptions are listed in Table 2 to facilitate the later description.

#### 4.3.1. Registration Phase

Every node in the network must register with the AAA to establish trust communication. Figure 3 shows registration procedures.

The 6L_MN_ sends a unique identification, *IDi*, and a randomly chosen random number, *R1*, to the AAA. As a result, a secure handshake is initiated, and each session begins with fresh inputs, thus minimizing the risk of replay attacks.Then, AAA picks up a random number, *R*_2_, and computes *Kr = H* (*ID_A_*)  ⊕
*R*_2_, where H represents a secure hash function, and is the AAA server’s unique identifier. The XOR operation ensures that the resulting *Kr* is tied to the server’s identity and the current session. To strengthen the protocol’s resilience, the session key, *Kr*, is divided into four equal chunks, each being 64 bits in size: *Kr1*, *Kr2*, *Kr3*, and *Kr4*. The secret parameter, *SP*, is then calculated as *SP = Kr1* ⊕ *Kr2* ⊕ *Kr3* ⊕ *Kr4*. This mechanism ensures that any modification to the key chunks will result in a different *SP*, which assists in detecting any unauthorized modifications. It acts as a dynamic and robust shared secret, ensuring message integrity and synchronizing cryptographic operations between the AAA server and the 6L_MN_. After this, the AAA server calculates a one-time alias identity, *AID_i_* = *H* (*ID_i_*||*R*_2_||*SP*), so that an attacker cannot link this alias identity with the actual *ID_i_* to correlate communications or track the 6L_MN_. *AID_i_* = {*AID1*, *AID2*, *…*, *AID_n_*}, where each *AID_n_* ∈ *AID_i_* is represented as a set of dynamically generated values for distinct sessions, ensuring privacy. The AAA server picks up the temporary ID (*TID_A_*) and sends the message to 6L_MN*i*_: {*AID_i_*||*TID_A_*||*R*_2_||*SP*}. SP acts as a shared secret between the AAA server and 6LoWPAN nodes, authenticating messages and ensuring synchronized cryptographic operations between authorized entities.Once both parties have stored the secret parameters in the database, a pairwise key is computed using *K_i_*↔*_A_* = *H* (*AID_i_*||*TID_A_*||*R*_2_||*SP*). The 6L_MN*i*_ publicly shares the *AID_i_* instead of utilizing the actual identity, hence thwarting the attacker’s ability to collect device information and keep track of the node.

**Figure 3 sensors-25-01458-f003:**
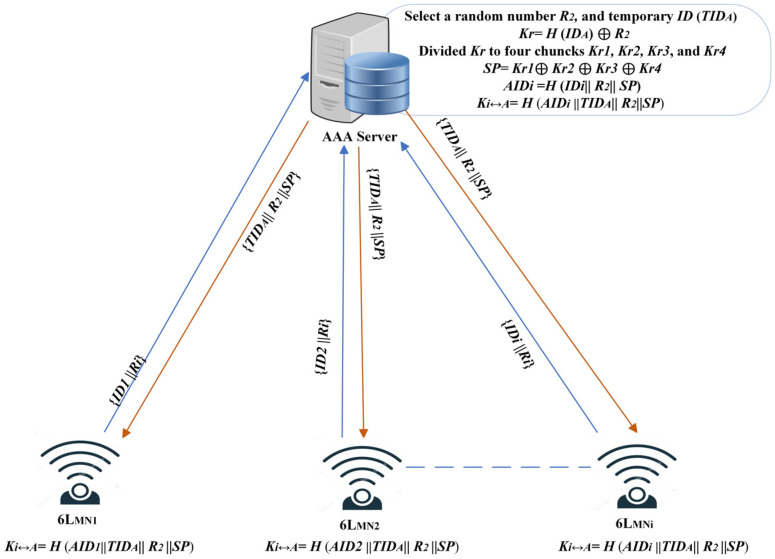
6L_MN_ registration with the AAA server provides all group members with temporary identification numbers, secret parameters, and random numbers. The key is divided into chunks, and the secure keys are distributed to the group members.

#### 4.3.2. Initial Group Authentication Phase

Upon the first joining of a group of registered 6L_MNis_ to the domine, it is necessary to authenticate with the server. During this stage, a set of 6L_MN*i*_, 6L_MAGs_, LMA, and AAA with each other will establish mutual initial AKE, as depicted in Figure 4 and outlined as follows:Upon a group of 6L_MNis_ joining the 6LoWPAN network, 6L_MNGL_ will be selected based on its capabilities. Every 6L_MNi_ of the group denoted as 6L_MN*i*_ 1 ≤ *i* ≤ *n* produces a *Tag_i_* = *MAC_ki_*↔*_A_* (*AID_i_*||*T_i_*||*R_i_*^2^), which is the tag in a standard MAC using the node pairwise key. Therefore, each group member sends its verification message {*AIDi*||*Ti*||*Tag_i_*} to the 6L_MNGL_ (FFD node).The 6L_MNGL_ collects all the authentication messages and then generates a 64-bit random number, *N_1_.* It then combines this number with its own authentication information, represented as *Tag_GL_* = *MAC_KGL_*↔*_A_* (*AID_GL_*||*T_GL_*||*N*_1_), including the messages from all the members of the group. 6L_MNGL_ incorporates all *M* group tags and combines its tag into *6GLM* = {(*AID*_1_, *Tag*_1_), …, (*AID_N_*, *Tag_N_*), (*AID_GL_*, *Tag_GL_*)}. Next, a router solicitation (RS) message is transmitted to the 6L_MAG_ as {*AID_GL_*||*T_GL_*||*MAC_KGL_*↔*_G_* (*6GLM*)}.When the message is received, the 6L_MAG_ encrypts and transmits the message {*ID_G_*||*T_G_*||*AID*_GL_||(*M*||*Tag_M_*)} to the AAA server. *M* = {(*AID*_1_, *Tag*_1_), …, (*AID_N_*, *Tag_N_*), (*AID_GL_*, *Tag_GL_*)}and aggregate *Tag_M_* = *Tag*_1_ ⊕ *Tag*_2_ ⊕…⊕ *Tag_N_* ⊕ *Tag_GL_*. Authentication, authorization, and session key distribution are the first steps that the LMA performs with the AAA server before engaging in mobility management and securely communicating with other entities, such as the 6L_MNGL_ and 6L_MAGS_. These steps are necessary before it can engage in mobility management. In this manner, a secure foundation can be established for managing handovers and maintaining the integrity of the network.Once the AAA obtains the message from the 6L_MAG_, it can decrypt the ciphertext using the key *K_i_*↔*_A_* associated with the *AID_GL_* (*AID_i_* before selected as GL) stored in the database. Upon receiving the Agg-MAC, the server ensures every tag using the pairwise key, *K_i_*↔*_A_*. This technique examines the Illegitimate node; if a 6L_MN*i*_ message is sent to the 6L_MAGs_ and the 6L_MNGL_ to notify them that this 6L_MN*i*_ is an unauthorized device, that should be terminated.Then, the AAA generates two nonces of 64 bits, *N*_2_ and *N*_3_, that will be transmitted to the 6L_MNGL_ for generating session keys with either the 6L_MAG_ or the LMA, as well as assigning an *ID_6G_* to each group member and a key calculating function for pairwise keys that are established between each group leader and its group member, *F*(*X*) = $\sum_{i=1}^{N}\left(K_{i↔GL}.\prod_{1\leq j\leq N,\;j\neq i}\frac{x-{AID}_{j}}{{AID}_{i}-{AID}_{j}}\right)\;$, used to compute the ciphertext (*CT_GL_*) and send it to 6L_MN*i*_ within a router advertisement message (RA). Then, AAA calculates a unique ticket of 64 bits in size, *T_C_* = (*ID_6G_* ⊕ *N*_2_ ⊕ *N*_3_ ⊕ *SP*), and *Texp* is the ticket expiry time of 64 bits of size used during the handover process. After that, a message {*T_C_*||*TID_A_*||*T_A_*||*Texp||E_KLMA_*↔*_A_*(*ID_6G_*||*N*_3_||*T_C_*||*Texp*||*ID*_*L**i**s**t*_||*CT*_1_, …, *CT_N_*, *CT_GL_*, *CT_G_*)} is transmitted to LMA, using *K_LMA_*↔*_A_* the secret shared key between the AAA and LMA, *ID*_*L**i**s**t*_ = *ID*_1_,…, *ID_GL_*, *AID*_1_, …, *AID_N_*, *AID_GL_*, *CT_i_* = *E_Ki_*↔*_A_*(*AID_GL_* ⊕ *H* (*AID_i_*, ${R_{2}}_{2}^{i}$) ⊕ *K_i_↔_GL_*||*ID_6G_*||*H*(*ID_6G_*||*K_i_↔_A_*||*T_C_*||*T_A_*)), an encrypted message or *CT_GL_* = *E_KGL_*↔*_A_*(*TID_A_*||*T_A_*||*T_C_*||*N*_2_||*N*_3_||*ID_6G_*, *f**x* ⊕ *H* (*AID_GL_*, ${R_{2}}_{GL}^{i}$), and an encrypted message for *CT_G_* = *E_KG_*↔*_A_*(*TID_A_*||*T_A_*||*ID_G_*||*N*_2_).Once the message is received from AAA, the LMA initially decrypts it and verifies its authenticity as originating from the AAA. If successful decryption is achieved using the *K_LMA_↔_A_*, *AID_i_* (where *i* = 1, …, *N*, including *AID_GL_*) can be obtained from *ID*_*L**i**s**t*_. Subsequently, the session key established between LMA and 6L_MNGL_ is calculated using *K_GL_*↔*_A_* = *H* (*AID_GL_*||*N*_3_||*ID_LMA_*||*ID_6G_*) based on *N*_3_. Hence, *AID_i_* and the temporary identity list are then stored in conjunction with the *ID_6G_* and *N*_3_ within the LMA database. Following this, home network prefixes (HNPs) are assigned by the LMA to each device, enabling it to configure its IP address accordingly. {*T_C_*||*Texp*||*ID_G_*||*T_G_*||*E_KG_*↔*_LMA_* (*ID_G_*||*T_G_*||*TID_6G_*||*HNP*_*L**i**s**t*_||*CT_G_*||*M*^X^||*CT_GL_*||*CT*_1_, …, *CT_N_*)} is transmit to the 6L_MAG_, where a temporary group identity is *TID_6G_* = *H* (*ID_6G_* ⊕ *N*_3_||*AID_GL_*||*T_A_*), *HNP*_*L**i**s**t*_ = *TID*_1_, …, *AID_GL_*, *HNP*_1_, …, *HNP_GL_* and the *M*^X^ = *E_KGL_*↔*_LMA_* (($\sum_{i=1}^{N}AIDi$ ⊕ $\sum_{i=1}^{N}HNPi$ ⊕ *AID _GL_* ⊕ *HNP_GL_*)||*ID_6G_*).Once the message is received from the LMA, the 6L_MAG_ decrypts it and ensures its validity. *N*_2_ is extracted from the ciphertext, *CT_G_*, using *K_G_*↔*_A_*. Subsequently, the session key between the 6L_MAG_ and the group leader is computed as *K_GL_*↔*_G_* = *H* (*AID_GL_*||*N*_2_||*ID_G_*||*TID_6G_*). It is possible to determine the *HNP* of each group member using the *HNP_List_*. Following that, *HNP*_*L**i**s**t*_, *M*^X^, *CT_GL_*, and *CT*_1_, …, *CT_N_*, will be sent to 6L_MNGL_ in a RA message {*T_C_||ID_G_*||*T_G_*||*CT_GL_*||*E_GL_*↔*_G_* (*ID_G_*||*T_G_*||*TID_6G_*||*HNP*_*L**i**s**t*_||*M*^X^||*CT*_1_, …, *CT*_N_)}.Upon receiving the RA message, 6L_MNGL_ decrypts *CT_GL_* using *K_GL_*↔*_A_*. This allows the 6L_MNGL_ to extract *N*_2_ and *N*_3_ from the ciphertext as *D_KGL_*↔*_A_* (*TID_A_*||*T_A_*||*T_C_*||*Texp*||*N*_2_||*N*_3_||*GID*, *f**x* ⊕ *H* (*AID_GL_*||${R_{2}}_{GL}^{2}$). The function *f**x* is stored in memory and used during pairwise key establishment with each group member. The key *K_GL_*↔*_LMA_* is derived using *N*_3_ by calculating *K_GL_*↔*_LMA_* = *H* (*AID_GL_*||*N*_3_||*ID_LMA_*||*ID_6G_*), and *K_GL_*↔*_G_* can be generated based on *N*_2_ using the calculation *K_GL_*↔*_G_* = *H* (*AID_GL_*||*N*_2_||*ID_G_*||*TID_6G_*). Following this, *HNP*_*L**i**s**t*_, *M^X^*, and *TID_6G_* are calculated from the encrypted message successfully. The *ID_6G_* and *TID_6G_* are verified by *ID_6G_* = *H* (*N*_3_||*AID_GL_*||*T_A_* ⊕ *TID_6G_*). Using the *HNP*_*L**i**s**t*_, *CT_i_*_,_ and *HNP_i_* are extracted for each group member. An authentication response message {*AID_GL_*||*T_GL_*||*CT_i_*||*T_C_*||*Texp*||*E_Ki_*↔*_GL_*(*AID_GL_*||*T_GL_*||*HNP_i_*) is forwarded to each 6L_MN*i*_ that is a group member after verifying the value *M^X^*, ensuring message integrity during transmission.


**Figure 4 sensors-25-01458-f004:**
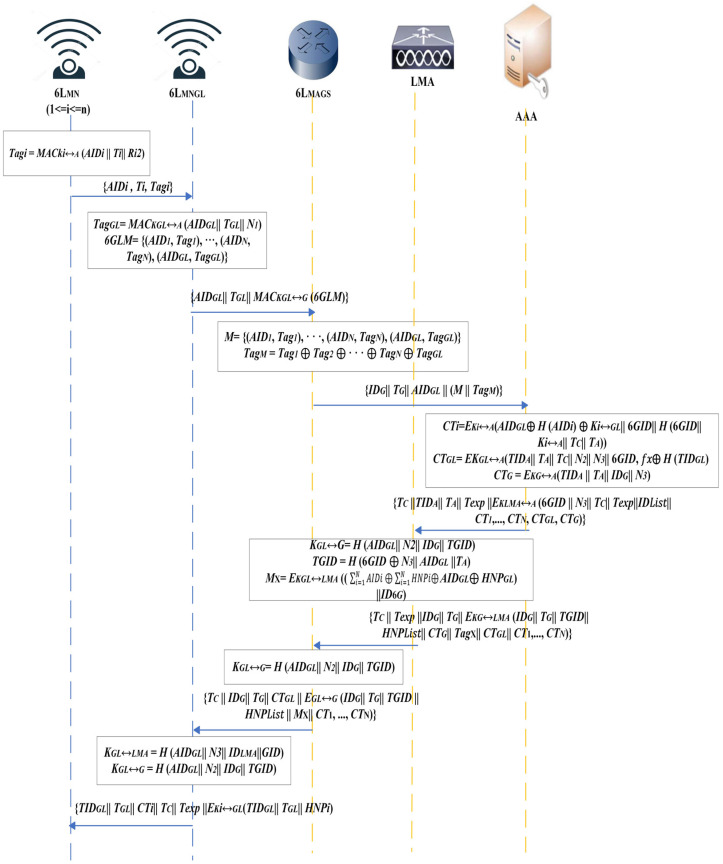
Initial authentication phase.

Once the authentication response message is received, each member decrypts *CT_i_* to verify *ID_6G_* and *K_i_↔_GL_*. This enables the recovery of *HNP_i_* from the encrypted message using *K_i_*↔*_GL_*. *T_C_* and *Texp* are stored in each 6L_MN*i*_ for use in the handover process.

#### 4.3.3. Group Handover Phase

Several related group handover schemes (GHSs) need authentication procedures to be performed each time a 6L_MN_ is attached to a new 6L_MAG_, increasing handover latency and workload. Through the use of ticket and secret parameter mechanisms, the proposed SL_GAS scheme facilitates secure group handover authentication while reducing handover delays. The group handover phase is shown in Figure 5.

In the event that the prior 6L_MAG_^old^ detects that a 6L_MNGL_ has left it as a domain, it will send to LMA a de-registration proxy binding update (De-PBU). This group of data packets is buffered.Prior to entering the domain of 6L_MAG_
^new^, the 6L_MNGL_ needs to collect *T_C_*’s *T_exp_* stored in its memory. If *T_C_* still has not expired, 6L_MNGL_ will pick *T_GL_* as the timestamp. Then, for each node, the handover information is collected as *Tagi* = *MAC_Ki↔A_* (*ID_6G_*, *AID_i_*, *T_i_*, and *R_2 i_
^j^*), where *j* denotes the *j*th handover of this group. When the 6L_MNGL_ attaches to a 6L_MAG_^new^, an RS message {*T_C_*||*AID_GL_*||*T_GL_*||*TID_6G_*||*CT_A_
*= *E_KGL_*↔*_A_* (*T_C_*||*AID_GL_*||*T_GL_*||*TID_6G_*||6*GLM*)} is dispatched, where the group handover information is *6GLM* = {(*AID*_1_, *Tag*_1_), …, (*AID_N_*, *Tag_N_*), (*AID_GL_*, *Tag_GL_*)}.Once the message is received, 6L_MAG_^new^ (*NG*) forwards a proxy binding update (PBU) message {*T_C_*||*T_NG_*||*ID_NG_*||*TID_6G_*||*CT_A_*||*M_NG_* = *E_KNG_*↔*_LMA_* (*ID_NG_*||*T_NG_*||*AID_GL_*||*TID_6G_*||*CT_A_*)} to inform the LMA that a group of 6L_MNs_ are currently roaming within its domain under the temporary group identifier *TID_6G_*. After verifying the *TID_6G_* value of the received PBU message, the LMA forwards a message {*T_C_*||*ID_LMA_*||*T_LMA_*||*AID_GL_*||*TID_6G_*||*CT_A_*||*M_LMA_* = *E_KLMA_*↔*_A_* (*ID_LMA_*||*T_LMA_*||*AID_GL_*||*TID_6G_*||*CT_A_*)} to the AAA for further verification.Upon receiving the verification information, the AAA server computes the ticket using data stored in its database. It calculates the *T_C_*’ = (*ID_6G_* ⊕ *N*_2_ ⊕ *N*_3_ ⊕ *SP*) to verify its consistency with the received *T_C_*. If the calculated *T_C_* differs from the received value, the AAA server sends a warning message to both the 6L_MAG_ ^new^ and the LMA, indicating potential illegality within the mobile group. If the values match, message {*TID_A_*||*T_A_*||*TID_6G_*||*CT_LMA_*||*CT_NG_*||*CT_GL_*} will be sent to the LMA, where *CT_LMA_* = *E_KLMA_*↔*_A_*(*TID_A_*||*T_A_*||*ID_6G_*), *CT_NG_* = *E_KNG_*↔*_A_* (*TID_A_*||*T_A_*||*TID_6G_*||*N*_4_), *CT_GL_
*= *E_KGL_*↔*_A_*(*TID_A_*||*T_A_*||*N*_4_ ⊕ *H* (*AID_GL_*, ${R_{2}}_{GL}^{j\;⊕1}$)), and *N*_4_ is a random number.If the ciphertext, *CT_LMA_*, is successfully decrypted, the LMA updates the binding cache entry to record the 6L_MAG_^new^. Subsequently, the LMA forwards a de-registration proxy binding acknowledgment (De-PBA) to 6L_MAG_^old^ (ON) and forwards a proxy binding acknowledgment (PBA) message {*ID_LMA_*||*T_LMA_*||*TID_6G_*||*HNP_List_*||*CT_NG_*||*CT_GL_*||*Tag′_LMA_*} to 6L_MAG_^new^, where *M′_LMA_* = *E_KNG_↔_LMA_* (*ID_LMA_*||*T_LMA_*||*TID_6G_*||*HNP_List_*||*CT_NG_*||*CT_GL_*). Each node is assigned an HNPList for continuous communication.Once the PBA message is received, the 6L_MAG_^new^ forwards *CT_GL_* in a RA message {*ID_NG_*||*T_NG_*||*TID_6G_*||*HNP*_*L**i**s**t*_||*CT_GL_*||*M′_NG_*} by the 6L_MAG_^new^ to the 6MN_GL_, where *K_GL_*↔*_NG_* = *H* (*AID_GL_*||*N*_4_||*ID_NG_*||*TID_6G_*) and *M′_NG_* = *E_KGL_*↔*_NG_* (*ID_NG_*||*T_NG_*||*TID_6G_*||*HNP_List_*||*CT_GL_*)}. When the 6L_MNGL_ receives the message, it computes the key *K_GL_*↔*_NG_* based on *N*_4_ derived from *CT_GL_*. In order to verify the integrity of the information received, *M’_NG_* is computed. It is the 6L_MNGL_’s responsibility to inform the other group members if the *HNPs* on the *HNP_List_* differ from the previous ones.It is possible then to send buffered data back to the LMA and forward them to the 6L_MAG_^new^. Data packets are reordered by the 6L_MAG_^new^ and delivered in sequence to the group leader. The group leader further transmits the data packets to the corresponding nodes according to their destination addresses. Thus, the group of 6L_MN_ successfully attaches to the 6L_MAG_^new^.

## 5. Security Analysis

SL_GAS security is examined from three perspectives in this section. The first step in the process is to informally analyze the security of the proposed solution against a variety of malicious attacks. SL_GAS is logically correct based on the SVO (Syverson–van Oorschot) logic. The final step in the formal verification is using Scyther simulation as a tool to verify the security functionality.

### 5.1. Informal Security Analysis

Confidentiality and integrity: As a result of SL_GAS, MIPv6 is able to ensure the integrity and confidentiality of data through a secure key agreement. Each authentication message is protected using MACs, session keys, *SK*, group public keys, *HF*, and *SP*. Each communication uses pre-shared parameters (*SPs*), a ticket (*T_C_*), and *K_x_*↔_y_ that are never transmitted in plaintext over the air interface, ensuring that session keys cannot be compromised and that the scheme functions with confidentiality and integrity.Mutual authentication: Section 5.2 and Section 5.3 demonstrate that SL_GAS is able to ensure mutual authentication. A Scyther was used to ensure that the GID could not be tampered with and that its confidentiality could be assured. As a result, after the initial and handover authentication protocols are completed, 6L_MN*i*_ will receive the *CT_i_* from the AAA server and use the pairwise key to determine the group ID for each group member. Aggregation _*MAC* with tag is used in the SL_GAS to ensure mutual authentication between the 6L_MN*i*_, 6L_MNGL_, 6L_MAG_, and AAA. After receiving a message containing (*M*, *tag_i_*), which is the aggregation of all messages 6L_MN*i*_, AAA checks the legality of the 6L_MN*i*_ by comparing the values in the message with the values received in the message. If the pair keys are considered valid, a warning message is transmitted to illegitimate nodes to inform the 6L_MAG_.Key leakage attack: In the SL_GAS scheme, all keys are resistant to key leakage. The SL_GAS scheme renews the pre-shared *K* (*x*, *y*) keys and the session key between all parties of MPIPv6 networks after an appropriate period. If the key leaks, the attack will be resolved within the key update interval to reduce the impact on the system. Using the SL_GAS scheme, we encrypt secret parameters with a secret identity using the hash function. After receiving MAC with a tag from 6L_MN*i*_, the 6L_MNGL_ determines only the correct 6L_MN*i*_. In this way, the key leakage can be prevented.Privacy protection (unlinkability and anonymity): The SL_GAS scheme uses alias one-time identity (*AID*) to ensure the anonymity of 6L_MN*i*_ and AAA. The real identity of 6L_MN*i*_ is concealed using secret parameters (*SPs*) and *AID* combined with *HF*. This identity is only known to the AAA and changes with each handover. Additionally, using random numbers in each message can make identifying two messages from the same group member difficult, thereby achieving anonymity and unlinkability.PFS: Forward key secrecy is crucial for any key agreement mechanism. Due to this property, even if the long-term key is compromised, it will be computationally impossible for a third party to predict future session keys. The SL_GAS scheme uses a new random number, *AID*, and a secret parameter (*SP*) to compute hashed session keys. Since the *SP* is known only to the AAA, an adversary cannot discover the *SP* or the pairwise key from the communication. Both the 6L_MN*i*_ device and the AAA generate fresh keys in each session using a random number and *SP* value, thus ensuring complete forward key security.Protocol attack resistance: The 6L_MNGL_ uses random numbers, *R_X_*, for the generation of MAC to ensure that the verification codes are fresh. *Replay attack:* SL_GAS resists the replay attack using random numbers, and timestamps are employed in each message with the pairwise key. An adversary might resend a previous handover request message from a legitimate 6L_MNGL_, but each SL_GAS message includes a timestamp. The receiver compares the received timestamp with the current time and ignores the message if the timestamp has expired. This mechanism ensures resistance to replay attacks. A variety of security mechanisms, including group public keys, shared keys, *HF*, session keys, and *SP*, are used to protect authentication messages, which are confidential and integral. These mechanisms are resistant to impersonation attacks and MITM attacks. *Impersonation attack:* The SL_GAS scheme ensures all legitimate devices are registered with the AAA before deployment. If an adversary tries to impersonate a node, the AAA will send a warning message to the LMA and 6L_MNGL_ if the secret information differs from what is stored in the database. This prevents impersonation attacks. *MITM:* In the SL_GAS scheme, an adversary cannot masquerade as a legitimate *ID_i_* to deceive 6L_MN*i*_ because an alias one-time identity and secret parameters, part of the shared session key, have been established through secure channels by the AAA. The adversary cannot obtain or modify the temporary session keys and thus cannot establish communication with 6L_MN*i*_. *Sybil attacks:* A Sybil attack involves forging multiple identities to disguise numerous non-existent devices. The goal is to deceive the 6L_MAG_ and LMA into believing these devices communicate within the domain, potentially consuming network resources and causing DoS attacks. It is necessary for each node to register with the AAA; therefore, nodes with fake identities or secret values cannot be authenticated. Additionally, SL_GAS supports group authentication, which helps mitigate DoS attacks. *DoS attacks:* An attacker may launch DoS attacks by impersonating a legitimate 6L_MN*i*_ and sending numerous fake requests to the 6L_MNGL_. The SL_GAS scheme uses a MAC tag to ensure the integrity of 6L_MN*i*_ without aggregating messages. This prevents DoS attacks, as sending a false MAC to invalidate the group verification will not succeed because only the MACs of two users are aggregated. Thus, the SL_GAS scheme effectively restrains DoS attacks. *Privileged insider attacks:* An insider attack occurs when a legal insider of a network steals confidential information or injects false information into the network. SL_GAS ensures the security of all messages by using a pairwise key between two parties. If the attacker does not possess the correct secret key, extracting the *SP* of other parties is difficult. Further, the 6L_MN*i*_, the LMA, and the AAA are the only parties with access to confidential information, such as the individual’s real identity. Due to the assumption that LMA and AAA are trustworthy, confidential information about each node cannot be leaked by either party. It is, therefore, impossible for an insider to deduce information from nodes other than what is sent to them.

### 5.2. Analysis of SVO Logic

Syverson and Orschot proposed SVO logic [27] as a method for analyzing security protocols that offer a number of advantages over BAN logic [28], GNY logic [29], AT logic [30], and VO logic [31]. The SVO logic can be used easily, is extremely extensible, and has clear semantics. SVO logic has several advantages, including its clear semantics, high expansion capabilities, and ease of use. The purpose of this section is to formally examine the authentication scheme’s mutual authentication using SVO logic. The SVO logic was used for informal analysis, verifying secrecy, untraceability, and PFS properties. Initially, we describe the SVO logic, followed by a discussion of the analysis process.

#### 5.2.1. Symbols

Listed below are the relevant notations and descriptions for the purpose of facilitating the following security proof.

The SVO logic has the following rules:

**Modus ponens (MP):** From *φ* and *φ* ⊃ *ψ* infer *ψ*. It means that if you know that φ is true and that *φ* implies *ψ* (denoted as *φ* ⊃ *ψ*), then you can infer that ψ is also true.

**Necessitation (Nec):** From ⊢
*φ* infer ⊢ *P* believes *φ*. This rule means that if φ can be proven (i.e., ⊢ *φ*) without assuming anything specific, it can infer that any principal *P* can believe *φ*. This is a method of moving from logical truths to logical beliefs.

**Believe axioms: BA1**: *P* believes *φ* ∧ *P* believes (*φ* ⊃ *ψ*) ⊃ *P* believes *ψ*. If *P* believes both *φ* and that *φ* implies *ψ*, then *P* must also believe *ψ*. This is an example of basic logical reasoning applied to beliefs.

**BA2:** *P* believes *φ* ⊃ *P* believes (*P* believes *φ*). This axiom aims to ensure that beliefs are “closed under belief”. In other words, if *P* believes something, then P also believes that they believe it.

**Source-association axioms: SA1:** *P* $\overset{K}{↔}$ *Q* ∧ *R* received { *X Q* } *K* ⊃ (*Q* ⟼ *X* ∧ *Q* ∋ *K*). Suppose *P* believes that *Q* shares a key *K* with *R*, and *R* receives a message encrypted with *K*; then, *Q* sends message X.

**SA2:** *P* $\overset{K}{\Leftrightarrow }$ *Q* ∧ *R* received {*X Q*} *K* ⊃ (*Q* ⟼ *X* ∧ *Q* ∋ *K*). It is a similar concept to the previous one, but it only works if there is a bidirectional or mutual key association (*P* + (*K*) *Q*), meaning both parties, *P* and *Q,* use the key *K*, and the message *X* originates from *Q*.

**Receiving axioms: R1:** *P* received (*X1*, …, *Xn*) ⊃ *P* received *Xi*. According to this axiom, if P receives a list of items (X1, …, Xn), P also receives each of the individual items Xi within the list.

**Seeing: S1:** *P* received *X* ⊃ *P* sees X. If *P* receives *X*, then *P* has access to or can observe *X*.

**S2:** *P* sees (*X1*, …, *Xn*) ⊃ *P* sees *Xi*. If *P* can observe a list of items, then *P* can observe any individual item on that list. It is similar to **R1** in terms of “seeing” rather than receiving.

**Saying: S3:** *P* said (*X1*, …, Xn) ⊃ *P* said *X1* ∧ *P* sees X1. If *P* said the list (*X1*, …, *Xn*), then *P* must have said *X1* and seen *X1*.

**S4:** *P* says (*X1*, …, *Xn*) ⊃ *P* says *X1* ∧ *P* said (*X1*, …, *Xn*). It follows that if *P* says the entire list (*X1*, …, *Xn*), *P* says *X1* and the entire list (*X1*, …, *Xn*). Message trustworthiness is established through this chain of “sayings”.

**Jurisdiction J1:** *P* controls φ ∧ *P* says φ ⊃ φ. If *P* controls *φ* and *P* says *φ*, then *φ* must be true. It is essential in information security, where an authority over certain information makes a declaration, and it is deemed true if that statement is made.

**Freshness F1:** fresh (*Xi*) ⊃ fresh (*X1*, …, *Xn*). In the case of a fresh Xi (e.g., a new, unused nonce), the entire list (X1, …, Xn) will also be considered fresh. A scheme can benefit from this feature whenever freshness needs to be preserved across multiple elements.

**Nonce-verification NV1:** fresh(X) ∧ *P* said X ⊃ *P* says X. When X is fresh, and P says X, then P says X now. This ensures that fresh data can be accepted as a part of current communications.

**Symmetric goodness of shared keys SK1:** Shared key (*K*, *P*, *Q*) ≡ shared key (*K*, *Q*, *P*). It simply illustrates the symmetry in the relationship between two shared keys. When *P* and *Q* share key *K*, it does not matter whether you refer to it as P sharing with *Q* or *Q* sharing with *P*. It is the same relationship.

#### 5.2.2. Group Handover Authentication Goals

The handover authentication scheme aims to establish an authentication scheme between 6L_MNGL_ and 6MAG^new^ to achieve mutual authentication within a domain. The AAA server facilitates establishing a trust relationship between LMA, 6L_MNGL_, and 6MAG^new^ and allows them to generate a shared key. This scheme aims to achieve the following objectives:

G1: 6L_MNGL_ believes 6MAG^new^ says (*ID_G_*, *T_G_*), 6MAG^new^ believes 6MAG^new^ says (*ID_G_*, *ID_LMA_*, *ID_A_*, *T_LMA_*).

G2: 6L_MNGL_ believes shared key (*_KG_*↔*_LG_*, 6MAG^new^, 6L_MNGL_), shared key (*_KG_*↔*_GL_*, 6MAG^new^, 6L_MNGL_).

G3: 6L_MNGL_ believes fresh (*_KGL_*↔*_G_*), 6MAG^new^ believes fresh (*_KG_*↔*_GL_*).

#### 5.2.3. Assumptions


P1: 6L_MN*i*_ believes *IDi*, *i* = 1, …, *N*, *GL.*P2: 6L_MN*i*_ believes *6GLM.*P3: 6L_MN*i*_ believes *TID_GL_.*P4: 6L_MN*i*_ believes fresh (*Ti*, *T_A_*, *T_NG_*, *T_LMA_*).P5: 6L_MN*i*_ believes (6L_MN*i*_ $\overset{R_{2}i}{\Leftrightarrow }$ AAA).P6: 6L_MN*i*_ believes (6L_MN*i*_ *_Ki_*↔*_A_* AAA).P7: 6L_MN*j*_ believes (6L_MN*j*_ *_Ki_*↔*_A_* 6L_MNGL_), *j* = 1, …, *N.*P8: 6L_MNGL_ believes AAA controls (6L_MNGL_
$\overset{R6}{\Leftrightarrow }$ AAA).P9: 6MAG^new^ believes (6MAG^new^ *_KG_*↔*_LMA_* LMA).P10: 6MAG^new^ believes *TID*^new^*_G_.*P11: 6MAG^new^ believes (6MAG^new^ *_KG_*↔*_A_* AAA).P12: 6MAG^new^ believes fresh (*T_GL_*, *T_G_*, *T_LMA_*).P13: 6MAG^new^ believes LMA controls *TID_G_*.P14: 6MAG^new^ believes 6L_MNGL_ controls (6L_MNGL_
$\overset{R6}{\Leftrightarrow }$ 6MAG^new^).P15: LMA believes *IDi*, *i* = 1, …, *N*, *GL.*P16: LMA believes *6GLM.*P17: LMA believes *TGID*.P18: LMA believes fresh (*T_G_*, *T_A_*_,_
*T_LMA_*).P19: LMA believes (LMA *_KLMA_*↔*_A_* AAA).P20: AAA believes *IDi.*P21: AAA believes *6GLM.*P22: AAA believes 6L_MN*i*_ controls (6L_MN*i*_ $\overset{R_{2}i}{\Leftrightarrow }$ AAA).P23: AAA believes (6L_MN*i*_ *_Ki_*↔*_A_* AAA).P24: AAA believes (6MAG^new^ *_KG_*↔*_A_* AAA).P25: AAA believes (LMA *_KLMA_*↔*_A_* AAA).P26: 6L_MNGL_ believes fresh (*T_GL_*, *T_G_*, *T_LMA_*).P27: AAA server believes fresh (*R*_2__,_ *N*_2_, *N*_3_, *N*_4_).P28: 6L_MN*i*_ believes 6L_MNGL_ received {*AIDi*||*Ti*||*Tag_i_*} ⊃ 6L_MNGL_ believes 6L_MN*i*_ received {*TID_GL_*||*T_GL_*||*CT*_i_||[(*TID_GL_*||*T_GL_*||*HNP*_i_)]*E_Ki_*↔*_GL_*).P29: 6L_MNGL_ believes 6MAG^new^ received {*TID_GL_*||*T_GL_*||*MAC_KGL_*↔*_A_ 6GLM*)} ⊃ 6MAG^new^ believes 6L_MNGL_ received {*TID_LMA_*||*T_LMA_*||*E_KG_*↔*_L_*(*TID_LMA_*||*T_LMA_*||*TGID*||*HNP*_*List*_||*CT_G_*||*M*’||*CT_GL_*||*CT*_1_, …,*CT_N_*)}.P30: 6MAG^new^ believes 6MN_LMA_ received {*TID_G_*||*T_G_*||*TID*_G_||*MAC _KGL_*↔*_A_* (*6GLM*)}⊃ 6MN_LMA_ believes 6MAG^new^ received {*TID_A_*||*T_A_*||*E_KGL_*↔*_A_*(*GID*||*N*_3_||*TC*||*Texp*||*ID*_*List*_||*CT*_1_, …, *CT_N_*, *CT_GL_*, *C_G_*)}.P31: 6MN_LMA_ believes 6L_MNGL_ received {*TID_G_*||*T_G_*||*TID_G_*||*MAC _KGL_*↔*_A_* (*6GLM*))}⊃ AAA believes 6MAG^new^ received {*TID_G_*||*T_G_*||*TID*_G_||*MAC_KGL_*↔*_A_* (*6GLM*))}.P32: 6L_MNGL_ believes (AAA says *TID_G_*||*T_G_*||*TID*_G_||*MAC _KGL_*↔*_A_* (*6GLM*))} ⊃ AAA believes (6MN*_i_* says (*TID_GL_*||*T_GL_*||*CT*_i_||[(*TID_GL_*||*T_GL_*||*HNP*_i_)]*E_Ki_*↔*_GL_*).


#### 5.2.4. Security Proof


**Establishing Trust Between Entities**


S1: AAA received {*T**I**D**_GL_*||*T**_GL_*||*MAC* (*6GLM*)} *_KGL_*↔*_A_.*The AAA server receives a message containing the necessary identifiers, timestamp, and a MAC value securely encrypted using the shared key, *K_GL_*↔*_A_*. This indicates that the initial communication between the group leader and the AAA server has been successfully initiated.From P5, SA1, and P3, we can obtain the following:S2: AAA believes 6L_MNGL_ says {*T**I**D**_GL_*||*T**_GL_*||*MAC _KGL_*↔*_A_*(*6GLM*)}.From S2, P4, and F1, we can obtain the following:S3: AAA believes fresh {*T**I**D**_GL_*||*T**_GL_*||*MAC _KGL_*↔*_A_*(*6GLM*)}.From S2, S3, and J1, we can obtain the following:S4: AAA believes 6L_MNGL_ received {*T**I**D**_GL_*||*T**_GL_*||*MAC _KGL_*↔*_A_*(*6GLM*)}.From S4, P6, and BA2, we can obtain the following:S5: AAA believes 6L_MNGL_ received *MAC* (*6GLM*); the AAA believes each corresponding 6MNi truly sends the handover.

**Step S2–S5:** Using principles of authentication theory, the AAA server deduces that the GL genuinely sent the received message. This assurance is built through the following:Trust in the cryptographic properties of the MAC.Validation of the timestamp to ensure message freshness.Assurance of integrity, as the MAC guarantees, is that the message has not been altered.

The AAA server thus believes the handover request originates from the legitimate group leader.


**Authenticating the Local Mobility Anchor (LMA)**


S6: LMA believes {*ID_LMA_*||*T**_LMA_*||*T**I**D**_GL_*||*TID_6G_*||*C**T**_A_*}_*KLMA*_↔*_A_.*S7: LMA believes AAA believes {*ID_LMA_*||*T**_LMA_*||*T**I**D**_GL_*||*T**G**I**D*||*C**T**_A_*}.From S7, P17, P16, and F1, we can obtain the following:S8: LMA believes fresh {*ID_LMA_*||*T**_LMA_*||*T**I**D**_GL_*||*TID_6G_*||*C**T**_A_*}.From S7, S8, and NV1, we can obtain the following:S9: LMA believes AAA believes {*ID_LMA_*||*T**_LMA_*||*T**I**D**_GL_*||*TID_6G_*||*C**T**_A_*}.

**Step S6–S9:** Similar logic is applied to the communication between the LMA and the AAA. By analyzing the timestamps, MAC values, and cryptographic keys used in their interaction, the LMA becomes convinced that the AAA is authentic and that the information exchanged is fresh and untampered. This step is critical because the LMA serves as a central anchor in managing mobility and must trust the AAA for further secure handover processes.


**Ensuring Handover Security Between MAGs**


S10: 6MAG^new^ believes {*T**G**I**D*||*T**_GL_*, *M*’*_LMA_*}.From S10, P11, BA2, we can obtain the following:S11: 6MAG^new^ believes LMA says *TID_6G_.*From S1, P16, and F1, we can obtain the following:S12: 6MAG^new^ believes fresh *TID_6G_.*From S11, S12, and NV1, we can obtain the following:S13: 6MAG^new^ believes LMA believes *TID_6G_.*From S13, J1, we can obtain the following:S14: 6MAG^new^ believes *TID_6G_.*S15: 6MAG^new^ says {*T**I**D**_LMA_*||*T**_LMA_*||*T**I**D**_GL_*||*TID_6G_*||*N*_4_} *_KG_*↔*_A_.*From S15, P9, and BP2, we can obtain the following:S16: 6MAG^new^ believes AAA says {*T**I**D**_LMA_*||*T**_LMA_*||*T**I**D**_GL_*||*TID_6G_*||*N*_4_}.From S16, P10, and F1, we can obtain the following:S17: 6MAG^new^ believes fresh {*T**I**D**_LMA_*||*T**_LMA_*||*T**I**D**_GL_*||*TID_6G_*||*N*_4_}.From S16, S17, and NV1, we can obtain the following:S18: 6MAG^new^ believes AAA believes {*T**I**D**_LMA_*||*T**_LMA_*||*T**I**D**_GL_*||*TID_6G_*||*N*_4_}.From S18, P12, and J1, we can obtain the following:S19: 6MAG^new^ believes *N_4_.*From S17 and F1, we can obtain the following:S20: 6MAG^new^ believes fresh *_KGL_*↔*_G_.*From S14, S19, P9, and P12, we can obtain the following:S21: 6MAG^new^ believes fresh *_KGL_*↔*_G (_N_4)_.*

**Step S10–S21:** The new Mobile Access Gateway, 6MAG^new^, verifies the legitimacy of the identifiers, timestamps, and MAC values provided during the handover process. It ensures the following:

The freshness of the key *K_GL_*↔*_G_* is validated through nonce and cryptographic principles.

The integrity and authenticity of the key exchange process.

This step ensures that 6MAG^new^ and the group leader securely establish the session key, *K_GL_*↔*_G_*, which will be used for secure communication in the new environment.

S22: 6L_MNGL_ believes {(*T**I**D**_A_*||*T**_A_*||*N*_4_ ⊕ *H*(*T**I**D**_GL_*, ${R_{2}}_{GL}^{j\;⊕1}$)} *_KGL_*↔*_A_*.


**Group Leader (GL) Verifying 6MAG^new^**


From S22, P6, and BP2, we can obtain the following:S23: 6L_MNGL_ believes AAA says {(*T**I**D**_A_*||*T**_A_*||*N*_4_ ⊕ *H*(*T**I**D**_GL_*, ${R_{2}}_{GL}^{j\;⊕1}$)}.From S23, P4, and F1, we can obtain the following:S24: 6L_MNGL_ believes fresh {(*T**I**D**_A_*||*T**_A_*||*N*_4_ ⊕ *H*(*T**I**D**_GL_*, ${R_{2}}_{GL}^{j\;⊕1}$)}.From S23, S24, and NV1, we can obtain the following:S25: 6L_MNGL_ believes fresh {(*T**I**D**_A_*||*T**_A_*||*N*_4_ ⊕ *H*(*T**I**D**_GL_*, ${R_{2}}_{GL}^{j\;⊕1}$)}.From S25, P8, and J1, we can obtain the following:S26: 6L_MNGL_ believes fresh *N_4_.*From S24, F1, we can obtain the following:S27: 6L_MNGL_ believes fresh *_KGL_*↔*_NG_.*S28: 6L_MNGL_ received *MAC’ _NG_.*S29: 6L_MNGL_ believes 6MAG^new^ believes *_KGL_*↔*_NG_.*From S29, we can obtain the following:S30: 6L_MNGL_ believes *_KGL_*↔*_NG_.*

**Step S22–S30:** The group leader verifies that the 6MAG^new^ has received and authenticated the MAC value, ensuring the shared key is genuine and fresh. This is achieved by the following:Verifying freshness using nonce N4 and timestamp *T*_A_.Ensuring integrity through the MAC value and hash computations.Through this process, both the group leader and MAG are assured of mutual authentication and commitment to the shared key.

Through this verification, the group leader and 6L_MAG_^new^ are assured that the shared key is genuine and that the MAC value sent by the 6L_MAG_^new^ is accurate. Based on (21) and (30), it may be concluded that the group leader and the 6L_MAG_^new^ are committed to the key *K_GL_*↔*_NG_*.

### 5.3. Formal Verification Based on Scyther Tool

The Scyther tool [32] was used to verify the proposed group handover authentication protocol. Verified schemes are primarily characterized by secrecy and authentication. Several security protocol elements can be expressed using Scyther’s Security Protocol Description Language (SPDL), including protocol definitions, role definitions, and data types. Scyther supports several robust security models, including DY and eCK, and several claims, such as Weakagree, Secret, Alive, Nisynch, and Niagree. These claims possess strong security properties. Among these are the ability to protect message confidentiality, detect MITM and replay attacks, guarantee the PFS of SL_GAS, and detect key leaks. Black-box verification is incorporated into Scyther. It is up to each role to determine whether it can achieve the security attribute or security goal. The attack output graph will indicate failure if the declared security attribute is not met. Otherwise, it will indicate OK. Using SPDL, 6L_MNGL_ aggregates all authentication information of group members from 6L_MN*i*_.

In the **initial authentication phase**, secure communications are established in five primary roles—6L_MAG_, 6L_MNGL_, 6L_MN*i*_, LMA, and AAA. The protocol was modeled for this verification with two group members (6L_MN*i*_) and their associated interactions. Scyther’s flexibility allows scaling the number of group members, though it may prolong the analysis time. As shown in the verification results (Figure 6), the Scyther tool verified all critical security claims for each role:6L_MN*i*_: Verified the secrecy of its alias identity (*AID_i_*), tag (*Tag_i_*), and its synchronization properties (Nisynch).6L_MNGL_: Verified the secrecy of the group leader’s temporary identity (*Tag_GL_*), aggregated authentication information (*AID_GL_*, *T_GL_*, and *R3*), and synchronization with other entities.6L_MAG_: Verified the secrecy and integrity of the aggregated tag (*Tag_G_*) and synchronization.AAA: Verified the secrecy of critical parameters, such as the temporary credentials (*T_C_*) and nonce (*N*_3_), and their liveness properties.LMA: Verified the secrecy of the shared group session key (*E**_KG_*↔*_L_*) and the temporary nonce (*N*_2_).

The results demonstrate that the initial authentication phase is resilient to parameter changes and robust against various security threats, such as unauthorized access, impersonation, and replay attacks. The verified claims ensure that the protocol’s design adheres to high-security standards, providing a solid foundation for secure group communication in the 6LoWPAN network. Therefore, it considers *GID* to be the group’s identity. After *GID* has been verified in Scyther, the whole group is safe if there are no vulnerabilities.

**Figure 6 sensors-25-01458-f006:**
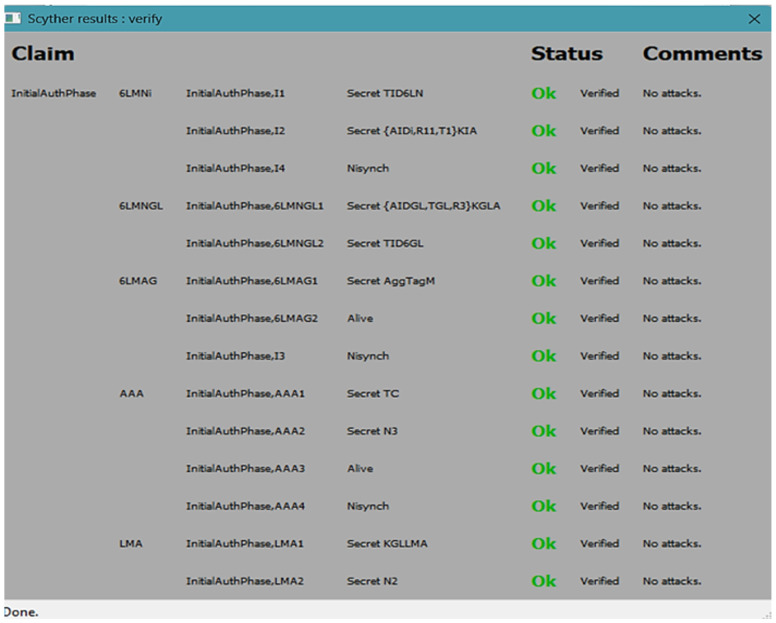
The results verify the initialization phase of SL_GAS.

At the SL_GAS’ **handover phase**, five basic roles are played: 6L_MAG_, 6L_MNGL_, 6L_MN*i*_, LMA, and AAA server. In this protocol, two members of the group are assumed. The number of group members in Scyther can be easily increased, although this may result in a prolonged analysis period. SPDL is used to code and analyze the proposed handover phase. SPDL is analyzed using Scyther. *TID_6G_* represents the temporary identity of the group, while the remaining roles correspond to the protocol entities.

As illustrated in Figure 7a–c, SL_GAS has been demonstrated to be resilient to parameter setting changes when tested with the Scyther tool. The three available parameters are the graph output, verification, and advanced parameters. The authors in [33] describe these parameters in detail. The following verification parameters were set up in the first setting: 50 runs, typed matching, 100 patterns for each claim, and 32 for the graph output. Finding the best attack is included in the advanced parameter; a search pruning procedure was set to locate the best attack, and the final graph output parameter was set to 32. In the advanced parameter, the component of the find best attack is represented, the search pruning parameter is represented, and the final graph output parameter is denoted by 32, as shown in Figure 7a.

In the second configuration, the modified SL_GAS verification parameters enable 100 runs to identify all issues, utilizing the matching type for identifying all issues, specifying a graph output parameter of 32, setting the search pruning parameter to locate the optimal attack, and employing 100 patterns per claim in the search pruning parameter, as shown in Figure 7b. In the third configuration, SL_GAS is used with the advanced search pruning parameters to identify all attacks; 100 patterns are allowed per claim, 32 graph output parameters are set, and a maximum of 100 verification runs is allowed for the verification component, as illustrated in Figure 7c. Consequently, no attacks were found on the SL_GAS by the Scyther tool under all advanced settings. Accordingly, A SL_GAS is capable of satisfying the security attributes detected by the Scyther tool, thereby proving its security.

## 6. Performance Evaluation

In this section, SL_GAS is compared with most related schemes [11,16,20], according to security requirements, computation cost, and transmission delay.

### 6.1. Security Analysis Comparison

According to the findings presented in Table 3, SL_GAS fulfills all the security criteria outlined in Section 3.3. It provides essential security features, such as PFS, protection against IP spoofing attacks, key leakage attacks, session unlinkability, and untrackability. Following each successful authentication, the secret key and other parameters, including identities, random values, and identifiers, are modified, resulting in added advantages for our suggested protocol. The SL_GAS authentication technique provides identity privacy preservation and safeguards against insider and traceability attacks, features that are not found in other authentication schemes, as seen in Refs. [11,16,20]. All the related schemes do not provide unlinkability, intractability, and PFS, and they cannot resist key leakage attacks. In addition, both schemes of Refs. [16,20] are vulnerable to Sybil and privileged insider attacks. None of them was created according to the CK or eCK-adversary models. In contrast, SL_GAS offers security in both the CK and eCK-adversary models, as well as SK security and verification. These features are not included in the authentication systems mentioned in Refs. [11,16,20]. Overall, SL_GAS is fortified against security breaches and is capable of fulfilling all security criteria.

### 6.2. Signaling Cost

A signaling cost calculation needs to be performed for 6L_MNi_ between gateway nodes based on the number of messages sent between them in order to assess the potential for congestion on the network [34]. The messages within the group are not included in the calculation. Table 4 presents the results of the signaling overheads; it can be seen that SL_GAS has a significantly lower signaling overhead than the related work, Refs. [11,16,20].

### 6.3. Computational Overhead

A unified configuration of the parameters listed in Table 5 facilitates comparison. We consider only the execution time of cryptographic operations, as shown in Table 6, for all comparison schemes according to [11]. The simulations are conducted on a computer with Intel(R) Core(TM) i5-3317U CPU @ 1.70 GHz, 8 GB RAM, and Windows 7 (64-bit). *T_h_*, *T_LiCi_*, *T_AES_*, and *T_HMAC_* represent the computational overhead associated with SHA-256, LiCi-128, AES-CBC-256, and (HMAC)-SHA-256, respectively.

SL_GAS demonstrates that it is more efficient than most related work GHSs that utilize other wireless technologies. According to [10], the computational overhead of related GHSs that involve *n* devices in their schemes, SGMS, SEIP, and CGM6, is summarized in Table 7.

Various GHSs are evaluated by unified cryptographic operations when n is the number of devices in a group. Despite appearing less computationally intensive, the SEIP scheme has fewer nodes, reducing the computation overhead. The SEIP protocol consumes more time than the SL_GAS protocol when dozens of 6LoWPAN devices roam in the same network. To demonstrate the high efficiency of the SL_GAS scheme, some GHSs rely on other wireless technologies compared to SL_GAS. A comparison of some GHS is shown in Table 7 and Figure 8 regarding computation overheads for SGMS, SEIP, and CGM6 based on n devices.

### 6.4. Transmission Overhead

A simulation message is composed of a PHY preamble of 8 bytes, a MAC header of 21 bytes, an IPv6 header of 6 bytes compressed, a fragment header of 3 bytes, the RA figure based on the frame format of 26 bytes [35], and the RA/RS header of 26 bytes [34]. An additional overhead is associated with the fast authentication in the handover header and the 22-byte RS header. As shown in Table 8, the total energy cost associated with fast authentication during a handover is compared. According to [36], transmission has a significantly higher energy cost than local computation. To prolong the lifespan of the 6LoWPAN network, it is necessary to minimize the number of messages. Based on Table 8, comparing SL_GAS to SGMS, SL_GAS achieves a reduction in energy consumption of 6.59% for groups of 10 nodes and 7.07% for groups of 50 nodes.

### 6.5. Discussion

This section compares SL_GAS with the related schemes (SGMS, SEIP, and CGM6) to highlight the advantages of SL_GAS concerning security, signaling costs, computational overhead, and transmission delay.

#### 6.5.1. Security Features

Regarding security capabilities, SL_GAS is superior to SGMS, SEIP, and CGM6 because it offers a complete range of security features. Several critical attacks against which it provides strong resistance, including key leakage, insider attacks, and traceability attacks. A significant difference between SL_GAS and other schemes is that SL_GAS provides PFS. To enhance protection against IP spoofing and Sybil attacks, the SL_GAS dynamically updates secret keys, identities, random values, and other parameters after each successful authentication. The scheme is also resistant to privileged insider attacks, one of the most common vulnerabilities in SEIP and CGM6. The compliance of SL_GAS with the CK and eCK adversary models for session key (SK) security reinforces its superiority over other schemes that fail to meet these advanced adversary model criteria. Thus, SL_GAS has a well-rounded design that anticipates a broader range of security threats. This contributes to its reliability in real-world applications, particularly those operating in sensitive or resource-constrained environments, such as 6LoWPAN.

#### 6.5.2. Signaling Cost

According to the comparison of signaling costs, SL_GAS generates a lower signaling overhead than the other schemes, mainly when a handover occurs within a handover. Table 4 shows that SL_GAS reduces message signals to *n* + 6, the lowest value among the schemes, resulting in lower network congestion. In environments where many devices are operated, minimizing the signaling overhead is crucial to avoid bottlenecks in the communication infrastructure. However, SL_GAS is designed to balance security and efficiency by optimizing cryptographic operations using LiCi-128, reducing authentication overhead with ticket-based mechanisms, and minimizing transmission overhead through aggregated MACs and optimized message exchanges. Compared to [16] (4*n* + 10), SL_GAS reduces signaling overhead by 40%, making it more suitable for large deployments. While [20] (2*n* + 2) grows rapidly with increasing *n*, SL_GAS remains manageable even in high-density environments. Thus, SL_GAS keeps inter-gateway communication efficient as the network expands, preventing excessive overhead that may compromise the performance of the network. This efficiency improves the network’s overall performance, especially under heavy load.

#### 6.5.3. Computational Overhead

SL_GAS exhibits remarkable efficiency in terms of computational overhead. The cryptographic operations in SL_GAS, especially using the LiCi block cipher (LiCi-128), are optimized for the limitations of devices based on 6LoWPAN. Although SEIP has a slight advantage in computational cost for smaller groups of nodes, SL_GAS scales more efficiently as the number of devices increases, making it preferable in larger networks. SL_GAS significantly reduces the execution time for cryptographic operations in comparison with SGMS, SEIP, and CGM6 while maintaining a balance between security and speed. A resource-constrained network in which computation and energy resources are limited is particularly in need of this efficiency. As seen in Table 6 and Table 7, as a result of the decreased computing overhead, the devices remain responsive, and the energy consumption remains within acceptable limits, resulting in a longer service life for network equipment. As *n* increases, SL_GAS scales better than [11,20], which exhibit a steeper rise in computational cost.

#### 6.5.4. Transmission Overhead

Transmission overhead for low-power networks like 6LoWPAN must be minimized, as energy consumption directly impacts device longevity. SL_GAS consumes the least energy among the compared schemes because its message size is reduced, and fewer messages are transmitted. It can be seen from Table 8 that the energy costs for SL_GAS are consistently lower than those of SGMS, regardless of whether the device group consists of 10 nodes or 50 nodes. As a result of this efficiency, 6LoWPAN networks can extend the life of battery-powered devices, as extending their life is a major concern. A significant benefit of SL_GAS is that it reduces the amount of data transmitted during communication exchanges, thereby reducing energy consumption. This advantage results in longer battery life for 6LoWPAN nodes and reduces the need to replace or recharge batteries frequently, which is particularly important when deployments are remote or difficult to reach. As a whole, SL_GAS has significant advantages over SGMS, SEIP, and CGM6 across various dimensions, including security, computational overhead, signaling costs, and transmission overhead. As a result of these advantages, SL_GAS is an ideal solution for secure, efficient, and sustainable group-based handovers in 6LoWPAN environments that require both security and resource optimization. For small networks (*n* = 10), SL_GAS consumes 6.59% less energy than SGMS. For larger networks (*n* = 50), SL_GAS saves 7.07% energy, proving its efficiency in high-scale deployments.

## 7. Conclusions

The aim of this study is to introduce a secure handover framework that utilizes PMIPv6 for a group of 6LoWPAN devices that are resource-constrained. A number of secret parameters are used in the registration phase to establish confidential and robust session keys, while tickets are used to validate the initial group membership, ensuring efficient and secure handovers between mobile nodes and group leaders. SL_GAS’s resilience to several security risks is confirmed by formal automated verification with the Scyther tool and SVO logic, while an informal examination shows its ability to withstand known attacks. When contrasted with most related frameworks, such as SGMS, SPAM, and SIEP, SL_GAS is reduced regarding signaling cost, transmission delay, and computation overhead, according to our mathematical analysis. The SL_GAS framework enables secure and fast handovers while protecting against malicious assaults. In addition to providing secure and efficient group handovers, SL_GAS also maintains a balance between security and privacy, making it a robust solution for managing 6LoWPAN mobility. Our future research will address privacy concerns related to 6LoWPAN group mobility management to prevent attackers from tracking mobile devices based on public information or transmitted messages between entities.

## Figures and Tables

**Figure 2 sensors-25-01458-f002:**
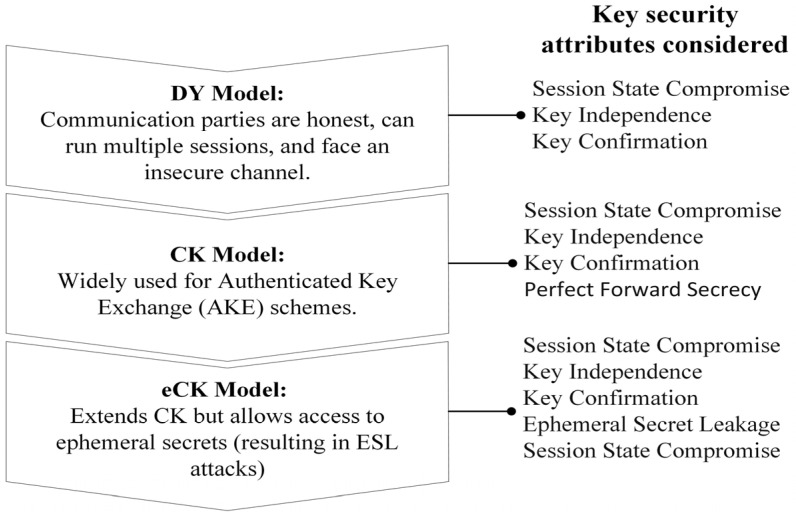
Key security attributes considered in DY, CK, and eCK adversary models.

**Figure 5 sensors-25-01458-f005:**
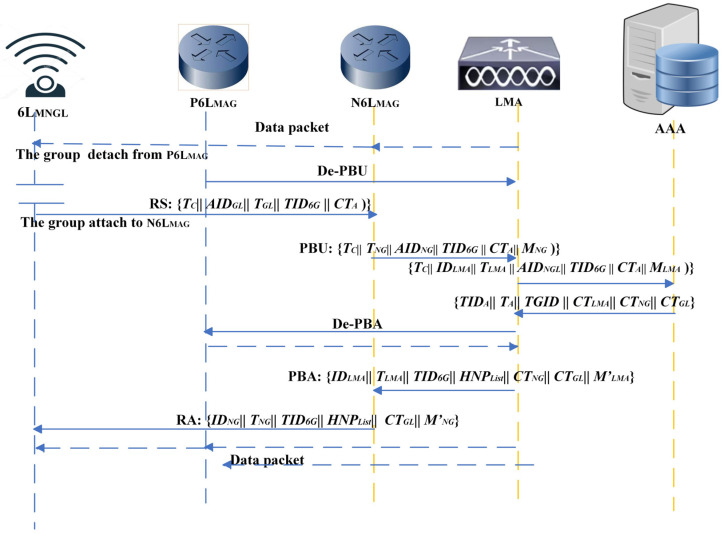
Group handover phase.

**Figure 7 sensors-25-01458-f007:**
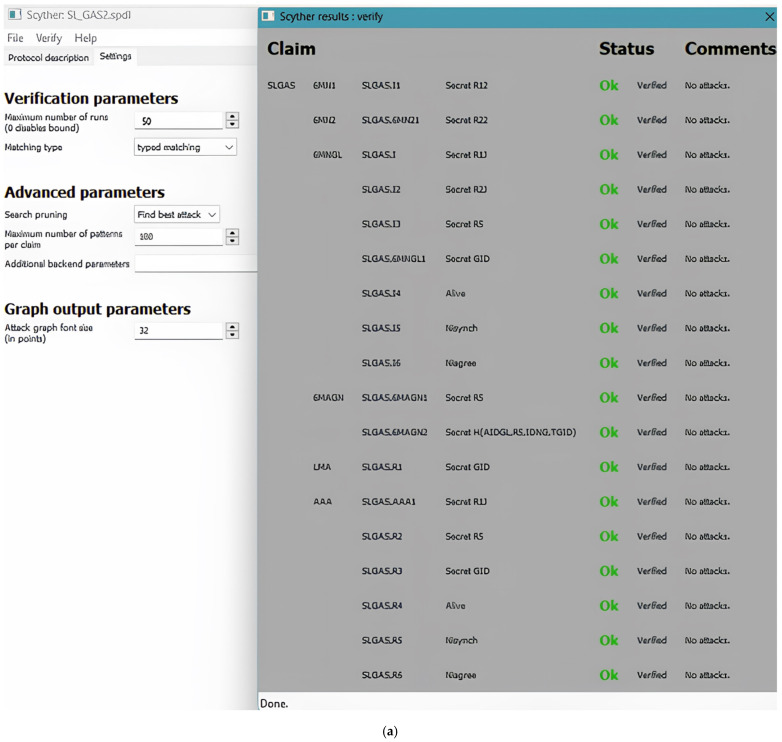
(**a**–**c**) Verify the results of SL_GAS under different settings in the Scyther tool. Remark: There has been an established level of security for these two main components of SL_GAS, namely aggregated MAC with tags and detecting functionality, as well as ensuring broadcast encryption. Formal verifications can be abstract terms. Thus, the emphasis is placed on the security of SL_GAS.

**Figure 8 sensors-25-01458-f008:**
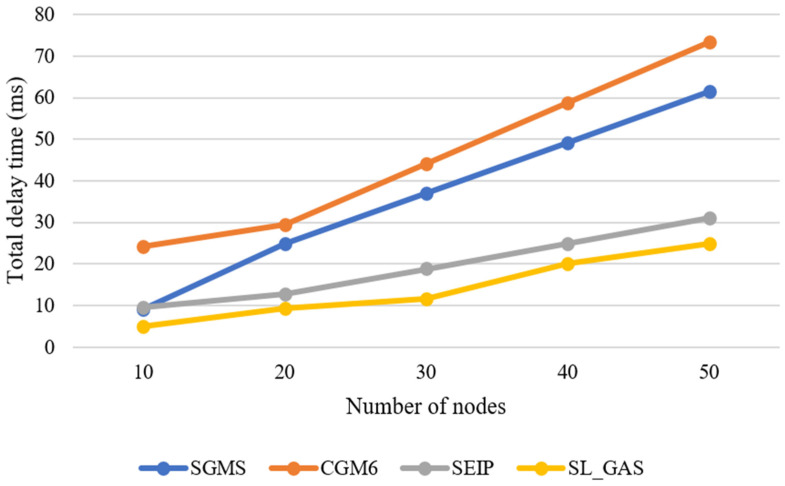
An analysis of various group-based handover schemes.

**Table 1 sensors-25-01458-t001:** Related-work summary.

References	Security Primitives	Security Properties	Security Issues (Limitations)	Security Proof Technique
Chen et al. [17]	Standard protocol	Not considered	Do not consider any attacks.	None
Wang and Mu [18]	Symmetric algorithm Temporary ID and hash function	Replay attack, address exhaustion attack, spoofing attack, and false address conflict attack,	Do not consider PFS, DoS, and leak key attacks.	Informal security analysis
SPAM [19]	Symmetric algorithm Temporary ID and hash function	Modification attack, stolen-verified attack, replay attack, and forgery attack.	Do not consider PFS, DoS, and leak key attacks.	Informal Security analysis
SGMS [11]	Aggregated MAC Symmetric algorithm Temporary ID and hash function	Replay attacks, MITM attacks, impersonation attacks, privileged insider attacks, and Sybil attacks.	Do not consider PFS and leak key attacks.	Informal Security analysis—BAN logic-Scyther tool
SEIP [16]	Aggregated MAC Temporary ID and hash function	Redirection and DoS attacks	Vulnerable to replay and MITM attacks; does not provide PFS	Informal security analysis
CBAS [20]	Aggregated MAC Temporary ID and hash function	Replay attack, MITM attack, impersonation attack, and modification attack.	Do not consider PFS, DoS, and leak key attacks.	Informal security analysis
Our proposed protocol (SL_GAS)	Aggregated MAC with tag Symmetric algorithm Ticket and secret parameters One-time alias and temporary ID and hash function	Impersonation attacks, replay attacks, privileged insider attacks, Sybil attacks, untraceability, PFS, session leakage key, and MITM attacks.	---	(Informal security analysis (DY, CK, and eCK adversary models)—SVO logic-Scyther tool)

**Table 2 sensors-25-01458-t002:** Notations.

Notations	Description
*A*	AAA server
*GL*	Group leader
*G*	6L_GM_
*ID_X_*, *TID_X_*	Identity and temporary identity of *x*
*AID_i_*	Alias identity of the 6L_MN_
*ID_6G_*, *TID_6G_*	The identity and temporary identity of the group
*SP*	Secret parameter
*Tc*	Ticket
*Texp*	Time for the expired ticket
*R_x_*	Random number of 64 bits
*N_x_*	Nonce
*K* _*x*↔*y*_	Established session key between *x* and *y*
*CT_x_*	Ciphertext for *x*
*H*	One-way hash function (*HF*)
*E_k_ (M)*	Encrypted message (*M*) with the key (*K*)
*D_k_ (CT)*	Decrypted ciphertext (*CT*) with the key (*K*)

**Table 3 sensors-25-01458-t003:** Comparison of security properties.

Features	[11]	[16]	[20]	SL_GAS
** *S1* **	Yes	Yes	Yes	Yes
** *S2* **	Yes	Yes	Yes	Yes
** *S3* **	No	No	No	Yes
** *S4* **	No	No	No	Yes
** *S5* **	No	No	No	Yes
** *S6* **	Yes	No	Yes	Yes
** *S7* **	Yes	Yes	Yes	Yes
** *S8* **	Yes	No	Yes	Yes
** *S9* **	Yes	No	No	Yes
** *S10* **	Yes	Yes	No	Yes
** *S11* **	Yes	No	No	Yes

Yes—indicates the availability of security functionality; No—indicates the security functionality is not applicable or is not taken into account. *S1*, confidentiality and integrity protection; *S2*, mutual authentication; *S3*, key leakage attack; *S4*, unlinkability and untraceability; *S5*, PFS; *S6*, replay attack; *S7*, impersonation attack; *S8*, MITM attack; *S9*, Sybil attack; *S10*, DoS attack; and *S11*, privileged insider attacks.

**Table 4 sensors-25-01458-t004:** Comparison of the signal cost.

Scheme	Intra-Handover
[11]	*n* + 7
[16]	4*n* + 10
[20]	2*n* + 2
SL_GAS	*n* + 6

**Table 5 sensors-25-01458-t005:** Parameter setting.

Parameters	Size (Byte)
Timestamp, nonce, HNP, *TID*, *ID*,	8
Key length	32

**Table 6 sensors-25-01458-t006:** The comparison of the execution time.

Cryptography Operations	(HMAC)-SHA-256	SHA-256	LiCi- 128	AES-CBC-256
Time (*ms*)	0.0617	0.0311	0.02	0.1253

**Table 7 sensors-25-01458-t007:** The comparison of the computational cost.

Schemes	Computational Overhead	Transmission Delay	Improvement vs. SL_GAS
Y. Qiu and M [11]	3*n*(*T_h_*) + (2*n* + 8) *T_HMAC_* + 8n(*T_AES_*) = 3*n*(0.0311) + (2*n* + 8)(0.0617)+ 8*n*(0.1253) = 1.2191*n* + 0.4936	*n*+ 9	82.2%
Lai et al. [16]	2*n*(*T_h_*) + (9*n* + 5) *T_HMAC_* = 2*n*(0.0311) + (9*n* + 5)(0.0617) = 0.06175*n* + 0.3085	4*n* + 10	64.9%
Imran et al. [20]	15*n*(*T_h_*) + 8*n*(*T_AES_*) = 15*n* (0.0311) + 8*n*(0.1253) = 0.4665*n*+ 1.0024*n* = 1.469*n*	9*n*	85.2%
SL_GAS	3*n*(*T_h_*) + (2*n*+ 2) *T_HMAC_* + 14*n*(*T_LiCi_*) = 3*n*(0.0311) + (2*n* + 2)(0.0617) + 14*n*(0.02) = 0.2167*n* + 0.4034	*n* + 7	*--*

**Table 8 sensors-25-01458-t008:** Energy cost comparison for authentication during handover.

Parameters	Energy Consumption (*Mj*)
SL_GAS for a group of 10 nodes	8.987
SL_GAS for a group of 50 nodes	36.876
SGMS for a group of 10 nodes	9.621
SGMS for a group of 50 nodes	39.682

## Data Availability

Data are contained within the article.

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
