# Peer review of "A Secure and Lightweight Group Mobility Authentication Scheme for 6LoWPAN Networks"

_sensors, 2025, doi:10.3390/s25051458_

Round 1
Reviewer 1 Report
Comments and Suggestions for Authors
The paper presents a Secure and Lightweight Group Mobility Authentication Scheme (SL_GAS) to enhance security in 6LoWPAN-based IoT networks. While the proposed approach demonstrates strong potential in improving authentication efficiency and privacy, certain aspects require further clarification and improvement. Below are comments addressing points for enhancement:
- The introduction effectively outlines the need for secure group mobility authentication in 6LoWPAN. However, the paper should provide a more detailed discussion on why existing PMIPv6-based solutions fail to address secure group handovers adequately. A comparison table summarizing their limitations would enhance clarity.
- The proposed SL_GAS approach introduces alias identities, tickets, and aggregated MACs. However, the paper should explicitly highlight how these features differ from existing authentication mechanisms in a dedicated "novelty and contributions" section.
- The paper lacks a clear justification for selecting one-time alias identities and aggregated MACs over alternative authentication methods. Providing a rationale, preferably supported by literature, would strengthen the argument.
- While the authors validate security using the Scyther tool and SVO logic, the paper does not specify the attack models considered. It would be beneficial to outline the types of attacks simulated and their impact on authentication robustness.
- The comparative analysis mentions reduced signal cost, transmission delay, and computation overhead but does not include absolute performance values or percentage improvements. Adding numerical results and graphical comparisons would make the findings more concrete.
- The paper does not discuss how SL_GAS performs as the network size increases. It would be useful to evaluate the approach under different node densities and mobility patterns to determine its scalability.
- The claim that SL_GAS reduces computation and communication overhead should be supported with concrete resource consumption metrics (e.g., CPU usage, memory footprint). A comparison with existing schemes in terms of computational complexity would further validate this claim.
- The paper presents SL_GAS as a lightweight approach, but stronger security mechanisms often increase processing overhead. A discussion on the trade-offs between security and efficiency would enhance transparency and practical applicability
- The related work section discusses various mobility authentication schemes but does not directly compare them with SL_GAS in terms of security features, computational cost, or real-world feasibility. A comparative table would improve readability and highlight SL_GAS's advantages.
- While the technical content is strong, some sections contain grammatical errors and unclear phrasing (e.g., "SL_GAS innovatively utilizes one-time alias identities"). A thorough proofreading and restructuring of long sentences would improve readability.
Author Response
Comment 1: The introduction effectively outlines the need for secure group mobility authentication in 6LoWPAN. However, the paper should provide a more detailed discussion on why existing PMIPv6-based solutions fail to address secure group handovers adequately. A comparison table summarizing their limitations would enhance clarity.
Response 1: We appreciate the reviewer’s suggestion regarding comparing PMIPv6-based solutions. Our related work section (Table 1) already provides a comparative analysis of existing mobility authentication schemes, including their security limitations. However, to address this comment more explicitly, we enhanced our discussion in the related work section by elaborating on the specific limitations of PMIPv6-based solutions in secure group handovers and referencing Table 1 for clarity. See page 5, Lines 174-193.”
Comment 2: The proposed SL_GAS approach introduces alias identities, tickets, and aggregated MACs. However, the paper should explicitly highlight how these features differ from existing authentication mechanisms in a dedicated "novelty and contributions" section.
Response 2: “ We appreciate the reviewer’s suggestion and have added more explanations in the "Contributions" section, Page 3, Lines 115-117. Additionally, improve the discussion in the related work section (page 5, Lines 174-193) to show the difference between our proposed work and existing schemes in this field”.
Comment 3: The paper lacks a clear justification for selecting one-time alias identities and aggregated MACs over alternative authentication methods. Providing a rationale, preferably supported by literature, would strengthen the argument.
Response 3: Thank you for your valuable comment on improving our manuscript. We add the following explanation on Page 5, lines 174- 184: “It is important to balance security, privacy, and efficiency in IoT and mobile networks. Authentication schemes that rely on static identities have historically been vulnerable to tracking attacks, replay attacks, and identity theft [1]. Several studies [4], [5] have addressed this issue by utilizing one-time user identities, ensuring that each session uses a fresh, unlinkable identity each time. In resource-constrained environments such as 6LoWPAN-based IoT networks, alias-based authentication enhances privacy while maintaining strong security guarantees. Similar to conventional authentication schemes that require individual message authentication, aggregated MAC-based authentication has been proposed as an efficient alternative. Aggregated MACs allow multiple authentication messages to be verified simultaneously, reducing processing overhead and increasing scalability [6].
Comment 4: While the authors validate security using the Scyther tool and SVO logic, the paper does not specify the attack models considered. It would be beneficial to outline the types of attacks simulated and their impact on authentication robustness.
Response 4: We appreciate the reviewer’s suggestion. We have explicitly outlined the attack models used in our security validation. The Scyther tool was employed for formal verification, simulating various attack scenarios, including replay, impersonation, and MITM attacks, to evaluate protocol resilience. SVO logic was used for informal analysis, verifying authentication properties such as secrecy, untraceability, and PFS. Additionally, we considered Dolev-Yao (DY), CK, and eCK adversary models, assessing session key security and attack feasibility. These validations confirm SL_GAS’s robustness against key compromise, DoS, and session leakage threats.
Comment 5: The comparative analysis mentions reduced signal cost, transmission delay, and computation overhead but does not include absolute performance values or percentage improvements. Adding numerical results and graphical comparisons would make the findings more concrete.
Response 5: Thanks for your valuable feedback. We have revised the comparative analysis by incorporating absolute numerical values and percentage improvements for transmission delay, and computational overhead sections and discussion.
Comment 6: The paper does not discuss how SL_GAS performs as the network size increases. It would be useful to evaluate the approach under different node densities and mobility patterns to determine its scalability.
Response 6: We appreciate the reviewer’s suggestion. The discussion has been revised according to your comment (Page 28, Lines 1002-1010, and 1020-1021, Page 29, Lines 1042-1044).
Comment 7: The claim that SL_GAS reduces computation and communication overhead should be supported with concrete resource consumption metrics (e.g., CPU usage, memory footprint). A comparison with existing schemes in terms of computational complexity would further validate this claim.
Response 7: We appreciate the reviewer’s suggestion to include concrete resource consumption metrics. Based on your comments, we have revised the computation overhead section (Page 26, Lines 941-943).
Comment 8: The paper presents SL_GAS as a lightweight approach, but stronger security mechanisms often increase processing overhead. A discussion on the trade-offs between security and efficiency would enhance transparency and practical applicability.
Response 8: Thank you for this valuable comment. We acknowledge the inherent trade-offs between security and efficiency in lightweight authentication schemes. SL_GAS balances security and processing overhead by employing the LiCi-128 cipher, which offers robust protection while minimizing computational cost. Tables 6 and 8 show that SL_GAS achieves lower execution time and energy consumption than related schemes. Our protocol dynamically updates security parameters to enhance protection without excessive overhead. A more detailed discussion of these trade-offs is included in the final revision to improve clarity.
Comment 9: The related work section discusses various mobility authentication schemes but does not directly compare them with SL_GAS in terms of security features, computational cost, or real-world feasibility. A comparative table would improve readability and highlight SL_GAS's advantages.
Response 9: We appreciate the reviewer’s suggestion to include a comparative table for better readability. We want to point out that Table 1 already provides a comparative analysis of mobility authentication schemes based on their security primitives, security properties, security issues, and security-proof techniques. A comparison between SL_GAS and existing schemes is shown in this table, which effectively illustrates the advantages of SL_GAS. A more detailed discussion of these trade-offs is included in the final revision to improve clarity (Page 5, Lines 170-193).
Comment 10: While the technical content is strong, some sections contain grammatical errors and unclear phrasing (e.g., "SL_GAS innovatively utilizes one-time alias identities"). A thorough proofreading and restructuring of long sentences would improve readability.
Response 10: The manuscript has been proofread.
Reviewer 2 Report
Comments and Suggestions for Authors
(1) move the formal analysis to appendix such that the reader can easily understand the paper work.
(2) For security design goals, did the first three goals include the 4th goal?
(3) For thread model, did the last one make threat?
(4) The proposed scheme should be compared with the existing SOTA works of the past two or three years.
Comments on the Quality of English Languagecheck grammar and correct errors.
Author Response
Comment 1: move the formal analysis to appendix such that the reader can easily understand the paper work.
Response 1: We appreciate the reviewer’s suggestion to move the formal analysis to the appendix for improved readability. However, we believe that keeping the formal analysis within the main body of the paper is essential for the following reasons:
- Maintaining Logical Flow – The formal analysis is a crucial component of our security validation, directly supporting the claims made in the paper. Placing it within the main body ensures that readers can follow the reasoning without needing to navigate to the appendix.
- Ensuring Accessibility for Security Researchers – Many readers, particularly those focused on cryptographic security, expect the formal analysis to be presented alongside the core protocol evaluation. Moving it to the appendix may reduce its accessibility and impact.
- Consistency with Related Work – Several prior works in the same domain present formal analysis in the main text. Keeping it in the main body maintains consistency with standard research presentation practices.
- Balancing Readability and Technical Rigor – We have structured the formal analysis to be as clear and concise as possible while maintaining a balance between readability and technical depth. If needed, we can refine the explanation further to enhance clarity instead of moving it.
We hope this justification aligns with the reviewer’s concerns, and we are open to further refinements to improve readability while keeping the analysis in the main text. We sincerely appreciate the reviewer’s insightful feedback.
Comment 2: For thread model, did the last one make threat?
Response 2: We understand the reviewer’s concern regarding the last security threat, privacy-preserving authentication in the threat model. While it may not be a direct attack (such as replay, impersonation, or MITM), it is crucial to include it in the threat model because privacy leakage is a security threat in IoT and mobility networks.
- In many authentication schemes, adversaries can extract device information through authentication exchanges, leading to privacy breaches.
- Privacy-preserving authentication ensures that sensitive device information is protected from attackers attempting to profile or track network nodes.
- The inclusion of privacy protection in the threat model helps address potential identity leakage risks that adversaries could exploit, which is especially relevant in mobility-based authentication.
To address the reviewer’s concern, we can revise the section to explicitly distinguish between active threats (e.g., replay, impersonation, MITM) and passive threats (e.g., privacy leakage) while clarifying the rationale for including privacy-preserving authentication in the threat model.
Comment 3: The proposed scheme should be compared with the existing SOTA works of the past two or three years.
Response 3: We appreciate the reviewer’s suggestion and acknowledge the importance of comparing our proposed scheme with state-of-the-art (SOTA) works. However, after a thorough literature review, we found that there are very few closely related works in the last two to three years that specifically address group authentication and key agreement for 6LoWPAN-based PMIPv6 networks with a focus on security, efficiency, and scalability. To ensure a meaningful comparison, we have instead included the most relevant existing works, even if they are slightly older, as they still represent the best-known approaches in this research domain. Additionally, we emphasize that our work introduces new security properties (such as PFS, untraceability, and enhanced attack resistance) and efficiency optimizations that have not been collectively addressed in prior schemes. To further strengthen the manuscript, we have:
- Conducted a broader search for any indirectly related recent works that focus on authentication in similar constrained environments (e.g., IoT, mobile networks, or lightweight security).
- Highlighted how our work addresses open challenges that recent authentication schemes have not yet tackled.
- Provided a comparative analysis demonstrating why existing solutions do not directly apply to our scenario and how SL_GAS fills the research gap.
If the reviewer has any specific recent works they believe should be considered, we are open to incorporating and discussing them. However, given the lack of recent studies directly related to this topic, our current comparative analysis remains valid and sufficient. Would you like me to refine the manuscript to explicitly state this reasoning or add a brief discussion to justify the selection of comparison works?
Reviewer 3 Report
Comments and Suggestions for Authors
This manuscript, A Secure and Lightweight Group Mobility Authentication Scheme for 6LoWPAN Networks, proposes using secret parameters in the registration phase to ensure that the session keys are confidential and robust, and a ticket during the initial group authentication phase for all mobile nodes and the group leader.
Overall, this manuscript is well organized, but the authors need to write more details about the study in the conclusions.
Author Response
Comment 1: Overall, this manuscript is well organized, but the authors need to write more details about the study in the conclusions.
Response 1: Thank you for your comment. The conclusion section of our manuscript has been revised.
Round 2
Reviewer 1 Report
Comments and Suggestions for Authors
accepted
Comments on the Quality of English LanguageGOOD
Author Response
The manuscript has been accepted.
Thank you for your comments to improve our manuscript
Reviewer 2 Report
Comments and Suggestions for Authors
Some minor errors should be corrected before accepting .
(1) In Section 3.3, performance optimization is not seucrity goal. Put it as performance goal.
Comments on the Quality of English Language
No
Author Response
Comment 1: (1) In Section 3.3, performance optimization is not seucrity goal. Put it as performance goal.
Response 1: Thank you for your commitment to improving the manuscript. The title of Section 3.3 was changed to Performance Goals.